# The Vitamin D Receptor as a Potential Target for the Treatment of Age-Related Neurodegenerative Diseases Such as Alzheimer’s and Parkinson’s Diseases: A Narrative Review

**DOI:** 10.3390/cells12040660

**Published:** 2023-02-19

**Authors:** Władysław Lasoń, Danuta Jantas, Monika Leśkiewicz, Magdalena Regulska, Agnieszka Basta-Kaim

**Affiliations:** Maj Institute of Pharmacology Polish Academy of Sciences, Department of Experimental Neuroendocrinology, Smętna 12 Str., 31-343 Krakow, Poland

**Keywords:** vitamin D, VDR polymorphisms, neuroinflammation, neuroprotection, neurodegenerative diseases

## Abstract

The vitamin D receptor (VDR) belongs to the nuclear receptor superfamily of transcription factors. The VDR is expressed in diverse brain regions and has been implicated in the neuroprotective, antiaging, prosurvival, and anti-inflammatory action of vitamin D. Accordingly, a relationship between vitamin D insufficiency and susceptibility to neurodegenerative diseases has been suggested. However, due to the multitargeted mechanisms of vitamin D and its often overlapping genomic and nongenomic effects, the role of the VDR in brain pathologies remains obscure. In this narrative review, we present progress in deciphering the molecular mechanism of nuclear VDR-mediated vitamin D effects on prosurvival and anti-inflammatory signaling pathway activity within the central nervous system. In line with the concept of the neurovascular unit in pathomechanisms of neurodegenerative diseases, a discussion of the role of the VDR in regulating the immune and vascular brain systems is also included. Next, we discuss the results of preclinical and clinical studies evaluating the significance of vitamin D status and the efficacy of vitamin D supplementation in the treatment of Parkinson’s and Alzheimer’s diseases, emphasizing the possible role of the VDR in these phenomena. Finally, the associations of some VDR polymorphisms with higher risks and severity of these neurodegenerative disorders are briefly summarized.

## 1. Introduction

The prevalence of chronic neurodegenerative diseases (NDs), such as Alzheimer’s disease (AD) or Parkinson’s disease (PD), is increasing as the population ages [1]. Moreover, it has been predicted that the SARS-CoV-2 pandemic could increase future neurological and neuropsychiatric complications [2,3]. Since SARS-CoV-2 infection evokes pronounced systemic inflammation linked with the promotion of cognitive decline and NDs [4], it is predicted that COVID-19 survivors could experience long-term increased neurodegeneration [5]. Thus, age-related NDs are considered the most important medical, social and economic problems of the contemporary world. The pathomechanism of neuronal damage may involve excitotoxicity, dysregulation of calcium homeostasis, perturbations in mitochondrial function, activation of proteolytic enzymes, increased production of reactive oxygen species (ROS) and a decreased antioxidant cell defense system. Deficiency of neurotrophic factors, pathological protein aggregation, cytoskeletal disruptions and neuroinflammation are also observed in the majority of NDs [6,7,8,9,10,11,12]. The abovementioned detrimental factors could be interconnected; one could evoke another, and some can appear simultaneously during the pathological process. Increased oxidative stress appears to play a predominant role in acute and chronic brain damage [13,14,15,16,17]. In the brain aging process, ROS overproduction is correlated with a reduction in the antioxidant defenses. Due to high metabolism connected with increased ATP expenditure, the brain possesses high oxygen demand (approx. 20% of the body’s oxygen). As it is also rich in fatty acids and has relatively poor antioxidant systems, this organ is particularly susceptible to oxidative damage. Moreover, neuronal cell damage in particular brain regions could be also facilitated by various transition metals (e.g., free iron) [18,19,20]. Neuronal loss observed in various acute and chronic neurodegenerative conditions could be executed by various forms of apoptotic (apoptosis) and non-apoptotic (necroptosis, ferroptosis, pyroptosis, autophagy) cell death [21,22,23], which are often inversely interconnected, causing some ambiguity when considering the sole inhibition of a particular path as a possible therapeutic strategy [24,25]. Accordingly, putative neuroprotective agents are sought among substances that, by inhibition of particular stages of neuronal cell death, such as ROS production, mitochondrial disruption, excitotoxicity, apoptosis, pyroptosis, necroptosis, ferroptosis or neuroinflammation, could delay the degeneration of neurons [12,26,27,28]. Since autophagy may play a dual role in neurodegeneration, its stimulation could be beneficial for age-related neurodegenerative conditions [29,30,31], whereas inhibition could be helpful in acute CNS injuries [32]. Because most of the preclinically studied putative neuroprotective drugs lack efficiency in clinical trials, attention has been placed on searching for substances with multitargeted action or combining drugs with distinct neuroprotective mechanisms. Moreover, such pleiotropic compounds could be beneficial for nonneuronal brain cells, such as astrocytes and microglia, which, as recently shown, can also actively participate in the neurodegeneration process though complex roles [33,34,35,36,37]. It should be stressed that targeting microglia and astrocytes could be a promising neuroprotective strategy since they are key regulators of inflammatory responses in the CNS [38,39,40,41,42,43,44]. Finally, the hypothesis of targeting neurovascular dysfunction as a protective strategy against acute and age-related NDs has been developed in recent years, although it remains poorly understood [45,46,47,48]. The growing awareness of the importance of structural and functional relationships between brain microvessels, astrocytes and neurons has led to the concept of the neurovascular unit. This unit builds the selective blood–brain barrier (BBB) to maintain brain homeostasis and control cerebral blood flow and comprises neurons, perivascular astrocytes, microglia, pericytes, endothelial cells and the basement membrane cells [49,50,51]. It has been hypothesized that chronic brain hypoperfusion, found in the early stages of various dementias, reduces oxygen delivery and lowers oxidative phosphorylation, decreasing the synthesis of adenosine triphosphate (ATP) and leading to oxidative stress, promoting inclusion bodies. Inclusions diminish the protective autophagic activity that removes malformed proteins and disrupt neuronal homeostasis [52]. Parkes et al. (2018) hypothesized that deciphering neurovascular crosstalk may offer novel therapeutic strategies for dementia and other NDs [53].

Vitamin D is a putative neuroprotectant, with multitargeting mechanisms of action that could affect neuronal and nonneuronal cell types and potentially target neurovascular dysfunction [54,55]. Vitamin D is involved in maintaining the homeostasis of the central nervous system (CNS) through vitamin D receptor (VDR)-mediated genomic effects and several other nongenomic mechanisms. Vitamin D has been reported to enhance neurotrophin gene expression and promote prosurvival and anti-inflammatory signaling pathways within the CNS. VDR and other nuclear hormone receptors also play a crucial role in myelination, promoting oligodendrocyte maturation and development and preventing demyelination processes [56,57].

In this review, we described the cellular and molecular mechanisms of vitamin-D-evoked neuroprotection, with special emphasis on its VDR-mediated genomic actions. We focused on the role of vitamin D in regulating neuronal cell death programs, which are involved in the pathogenesis of various neurodegenerative conditions. Moreover, we also summarized VDR-mediated vitamin D modulatory effects on astrocyte and microglia function and their impact on neuroinflammatory processes. Next, we consolidated the results of preclinical and clinical studies evaluating the status and efficacy of vitamin D in treating AD and PD, emphasizing the putative role of the VDR in these phenomena. Finally, the associations of some functional VDR single-nucleotide polymorphisms with higher risks and severity of these NDs are briefly described.

### Vitamin D and Molecular Characteristics of Its Genomic and Nongenomic Action

Vitamin D is a fat-soluble substance belonging to the secosteroid family of which five forms (D1, D2, D3, D4 and D5) have been described to date. In humans, the main source (approx. 80%) of vitamin D derives from the skin, where under sun exposure 7-dehydrocholesterol is transformed to unstable pre-vitamin D3 which further isomerizes to vitamin D3 (cholecalciferol). This form of vitamin D can also be consumed via animal-based food, whereas in plants, vitamin D2 (ergocalciferol) is present [55,58]. The third source of vitamin D includes commercially available supplements in which cholecalciferol predominates as the compound with better bioavailability and biological activity when compared to ergocalciferol. To achieve their biological effects, these inactive forms of vitamin D in the first step need to be metabolized in the liver by CYP2R1 (25-hydroxylase), where 25-hydroxyvitamin D3 (25(OH)D3, calcifediol) or 25-hydroxyvitamin D2 (25(OH)D2) are produced. These circulating vitamin D metabolites are further hydroxylated by CYP27B1 (1 α-hydroxylase) present in target organs (e.g., the kidney) which produce 1α,25-dihydroxyvitamin D3 (1,25(OH)2D3, calcitriol) or 1α,25-dihydroxyvitamin D2 (1,25(OH)2D2), the active forms of vitamin D3 and D2, respectively. Since calcitriol maintains calcium and phosphate bone homeostasis, its production and metabolism in the kidney are precisely regulated. Only about 0.4% of active or circulating forms of vitamin D metabolites are available in plasma in free form, whereas 58% are associated with vitamin D binding proteins (DBPs), which are highly polymorphic in humans, causing interpersonal differences in vitamin D bioavailability. When the levels of active and circulating (storage) forms of vitamin D in the body are too high, especially after excessive supplementation, they are metabolized by CYP24A1 to inactive metabolites (1,24,25(OH)3D, calcitroic acid and 24,25(OH)2D).

The VDR belongs to the nuclear receptor superfamily of transcription factors, which also includes corticoid, retinoid, thyroid and sex steroid hormone receptors. Depending on the target gene, the VDR ligands can stimulate or inhibit transcription [59]. The *VDR* gene is localized on chromosome 12q13, contains 14 exons and covers 75 kb of genomic DNA. Upon ligand binding (e.g., calcitriol), the VDR forms a heterodimer with the retinoid X receptor (RXR) and binds to vitamin D response elements (VDREs) in target genes to modulate their expression [60]. There are marked differences in the mechanisms of nuclear import of RXR and the VDR. RXR binds to importin beta, and its ligand, 9-cis-retinoic acid, only slightly increases the nuclear localization of this receptor. Translocation of the VDR to the nucleus is mediated by importin alpha, and this process is greatly enhanced by calcitriol. Moreover, upon binding with its ligand, the VDR plays a dominant role in mediating the nuclear import of RXR-VDR heterodimers [61]. Alterations in gene function and decreased affinity for calcitriol are frequently associated with single-nucleotide polymorphisms (SNPs) in the *VDR* gene. Four variants of the *VDR* gene have been described: FokI (rs10735810), TaqI (rs731236), BsmI (rs1544410) and ApaI (rs7975232), where the first two are localized in exonic regions, while the latter two are found in intronic regions of the *VDR* gene. These polymorphisms are associated with an altered protein function (TaqI) or expression (BsmI and ApaI) or altered translation initiation site (FokI) [62]. An active vitamin D metabolite directly and/or indirectly regulates approximately 3% of the mouse or human genome, as was estimated by Bouillon et al. (2008) [63], and almost all cells respond to its stimulation via the VDR. Nuclear VDR, for example, regulates enzymes involved in the endogenous synthesis and metabolism of vitamin D3 or genes responsible for the maintenance of bone calcium homeostasis. Moreover, the functions of the VDR and active form of vitamin D do not fully overlap, as evidenced by the development of total alopecia in VDR-deficient mice but not in vitamin-D- or CYP27B1-deficient mice [63]. It has been shown that when calcitriol level is low, the VDR can bind its low-affinity ligands, e.g., curcumin, polyunsaturated fatty acids or anthocyanidins, whereas some other factors such as resveratrol and sirtuin 1 can increase nuclear VDR signaling [64,65].

The rapid action of active vitamin D metabolites can be mediated by cell-membrane-localized VDR, which can modulate the activity of various signaling pathways (e.g., WNT, sonic hedgehog (Shh) and NOTCH signaling) or other membrane receptors (e.g., calcium channels or mitochondrial permeability transition pore). The enzyme PDIA3 (protein disulfide isomerase family A member 3, ERp57, 1,25D3-MARRS) is the best described target responsible for non-genomic action of active vitamin D metabolites [63]. By these mechanisms, they can regulate extracellular Ca2+ influx through L-VGCC and activate various protein kinases (e.g., CaMKII, PKC, PKA, PI3-K or MAPKs) and phospolipases (PLA2, PLC). Moreover, after binding with membrane receptors (VDR or PDIA3), active vitamin D metabolites can interact with some transcription factors (e.g., STAT3, NF-κB, Nrf2, RORα, RORγ, AhR) and in this way, they indirectly influence the expression of various genes. In contrast to the genomic action of vitamin D via VDR, its non-genomic mechanisms are less recognized and mostly evidenced in in vitro settings [55,58,63] (Figure 1). 

## 2. The Brain Expression of the VDR

The VDR is ubiquitously expressed. Immunohistochemical analysis has indicated the presence of the VDR in both human and animal brain structures, such as the hypothalamus, hippocampal formation, amygdala, stria terminalis and neocortex [66,67]. These findings were also confirmed by in situ hybridization, autoradiography and RT-PCR methods, which showed VDR expression in the adult human brain in the cortex, thalamus, and amygdala, throughout the olfactory system and hippocampus, and abundantly in the hypothalamus and substantia nigra [67,68,69,70]. Eyles et al. (2005) [71] found that the pattern of VDR distribution in the human and rat brain is very similar. The presence of the VDR was identified in neuronal and glial cells, with the highest VDR immunoreactivity in hippocampal CA1 and CA2 pyramidal cells; with a few exceptions, there was tight overlap between the VDR and CYP27B1, suggesting local brain calcitriol production. Wang et al. (2012) indicated in their review that early ELISA and immunohistochemical studies showed undetectable levels of VDR in the cerebrum and cerebellum of rats and humans, but also pointed out that the antibodies used for a direct measurement of VDR were not VDR-specific [71]. Recent imunohistochemical studies have shown the presence of VDR in the cortex, caudate putamen, amygdala and reticular thalamus, and to a lesser extent in the hypothalamus, hippocampus, dorsal raphe nucleus, paraventricular thalamic nuclei and bed nucleus of the stria terminalis in the brain of the VDRCre mouse line [72]. A comparative study on VDR and Pdia3 transcript expression measured by qPCR in cultured astrocytes, endothelial cells, microglia, neurons and oligodendrocytes derived from rats showed that the expression of *Vdr* in brain cells was less abundant than that in typical vitamin D target organs (kidney, liver), whereas *Pdia3* had the opposite expression. Among the tested various brain cell phenotypes, the highest expression of *Vdr* was found in astrocytes when *Pdia3* was abundantly expressed in neurons, astrocytes and endothelial cells [67]. Thus, astrocytes were suggested to be the most likely cell type to respond to calcitriol at the genomic level, whereas other cell types could be responsible for its rapid non-genomic effects [67]. It is uncertain whether VDR expression in the brain depends on the concentration of calcitriol. An in situ hybridization study showed that this ligand increased the pool of transcripts encoding the VDR, as well as nerve growth factor (NGF) and its low-affinity receptor, the p75(NTR) protein, in rat oligodendrocytes [56]. However, in a study by Landel et al. (2017), calcitriol addition to the culture medium produced a significant increase in the transcript coding for VDR in rat microglia but not in other types of brain cells (neurons, astrocytes, endothelial cells, oligodendrocytes) [68]. Another study showed that the presence of calcitriol is required for glutamate-induced *VDR* mRNA expression in cortical neurons [70]. Although prenatal hypovitaminosis D dysregulated synaptic plasticity and neurotransmission, producing profound changes in the adult brain and behaviors, it had no effect on transcript expression of *VDR* in the rat brain [72,73]. Inflammation may affect the expression of the VDR in the brain. A mouse lupus erythematosus model demonstrated increased IgG infiltration into the hippocampus, indicating that an inflammatory process enhanced VDR expression in this brain region [74]. Vitamin-D-induced VDR expression promoted microglial polarization toward the anti-inflammatory phenotype and subsequent neuroinflammation alleviation by suppressing the TLR4/MyD88/NF-*κ*B signaling pathway following traumatic brain injury (TBI) and contributing to the restoration of post-traumatic secondary brain injury in rats [75].

### Brain Aging

Aging is a highly complex biological process, in which the gradual impairment of the antioxidant defense system, deterioration of immune system activity (immunosenescence), and decline in metabolic and regenerative processes take place. It has been postulated that the age-related increase in ROS generation, along with dysfunctional cellular repair or degradation mechanisms, can lead to elevated level of oxidized proteins and formation of amyloids [76]. Moreover, it has been suggested that the pathogenesis of NDs, such as AD and PD, is associated with the well-recognized hallmarks of aging: cellular senescence, loss of proteostasis, deregulated nutrient sensing, mitochondrial dysfunction, telomere attrition, genomic instability, epigenetic alterations, altered intercellular communication and stem cell exhaustion [1]. Recently proposed new hallmarks of aging include autophagy, microbiome disturbance, altered mechanical properties, splicing dysregulation and inflammation [77]. Accumulating evidence indicates that vitamin D, via both its genomic and non-genomic actions, may influence aging processes. VDR stimulation increases neurotrophin gene expression and promotes prosurvival and anti-inflammatory signaling pathways within the CNS. Vitamin D and its derivatives have been evidenced to protect neuronal cells against a variety of toxic agents at nanomolar concentrations and to prevent premature aging. In conjunction with the pivotal role of oxidative stress in the pathomechanism of brain aging and neuro-degeneration, it was reported that the combined treatment with calcitriol and the powerful antioxidant lipoic acid protected primary mouse astrocytes against hydrogen-peroxide-induced apoptosis and iron-accumulation-induced damage [78]. A recent study demonstrated that the VDR is involved in attenuating ischemia-induced oxidative stress and brain injury via reciprocal activation of SMAD family member 3 (SMAD3) and VDR transcription factors [77]. Berridge (2017) [79] put forward a hypothesis that vitamin D, due to its pleiotropic mechanisms such as its effects on calcium homeostasis, controls the rate of aging, and that its deficiency may increase the probability of age-related diseases, including PD and AD. Indeed, some preclinical data indicate that vitamin D3 can affect the biological hallmarks of aging. Thus, the administration of a food supplement containing *Centella asiatica* (L.) extract, vitamin C, zinc and cholecalciferol for 3 months restored telomerase reverse transcriptase expression and enhanced telomerase activity in the cerebellum and cortex of 18-month-old rats [80]. Of note, calcitriol has been shown to stabilize 53BP1 and promote DNA double-strand break repair through the inhibition of CTSL, the upregulation of which is a hallmark of premature aging [81] [Gonzalez-Suarez et al., 2011]. As reviewed by Ashapkin et al. (2019) [82], there is some experimental evidence that vitamin D3 reduces progerin production and ameliorates progerin-related premature aging symptoms. In addition, an interaction of vitamin D with the ”aging suppressor” protein Klotho has been considered. Klotho shows pleiotropic biological effects and its insufficiency appears to play a role in premature aging, whereas its overexpression promotes longevity. Specifically, deficiency in the vitamin D/Klotho/Nrf2 regulatory network may promote age-related cognition deficits in rats, which can be attenuated by vitamin D supplementation [83]. Klotho is expressed in the brain, and besides serving a plethora of biological functions, it also regulates vitamin D metabolism [84]. Tuohimaa conducted studies on FGF23-/- and Klotho-/-mice and concluded that aging shows a U-shaped dependency on vitamin D levels, and that an optimal concentration of calcidiol to a higher extent than calcitriol is required to delay the aging process [85]. It has been demonstrated that VDR KO mice showed premature aging features, such as alopecia, thickened skin and epidermal cysts, although no changes in the number of Purkinje cells in the cerebellum were found. Moreover, it has been shown that the expression of genes encoding NF-kB, Fgf-23, p53 and IGF1R was significantly lower in old VDR KO mice as compared to the controls. The authors pointed out that mice with D3 hypervitaminosis and aged VDR knockout mice are phenotypically similar, and that genetic ablation of VDR promotes premature aging [86]. Messing et al. (2013) [87] showed for the first time that exposure to vitamin D3 increased the lifespan of wild-type Caenorhabditis elegans, which expresses DAF-12 nuclear hormone receptor homologous to the vitamin D receptor in humans. This finding was confirmed by Mark et al. (2016) [88], who also showed in a mechanistic study that cholecalciferol improved protein homeostasis and extended the lifespan of these worms via stress response pathway genes skn-1, ire-1 and xbp-1, and prevented protein aggregation and β-amyloid-induced toxicity. Vitamin D3 via immune regulation may also slow down the aging process. Thus, in aged mice, vitamin D3 administration for 6 weeks reduced hallmarks of aging, such as retinal inflammation, retinal macrophage numbers and amyloid beta accumulation [89]. Other investigators have suggested that a combination of calcitriol and mesozeaxathin could be a useful approach to combat pathological features of Age-Related Macular Degeneration (AMD), such as retinal inflammation and oxidative stress [90]. The anti-inflammatory effects of vitamin D3 administration in the retina may be relevant to therapies of neurodegenerative diseases because visual abnormalities due to AMD and glaucoma are often associated with AD and may occur before cognitive deficits [91]. Additional evidence has been provided that vitamin D3 can act as an anti-inflammatory agent by reducing age-related microglial activation and increasing IL-1beta level in the rat hippocampus [92]. Collectively, preclinical data suggest that vitamin D3 interferes with key regulators of aging processes and that a lack of VDR promotes progeria. However, the optimal vitamin D serum concentration for slowing down brain aging has not been established yet. Peng et al. (2021) [93] reported an increase in Aβ accumulation in the mouse brain during aging, which was associated with the n P-glycoprotein (P-gp) level. Calcitriol acting via VDR prevented age-related changes in P-gp level and partially reduced Aβ brain aggregates. Khairy and Attia (2021) [94] investigated the possible protective effects of oral supplementation of cholecalciferol in young, middle-aged and old rats, and found that vitamin D significantly mitigated the aging-related reduction in brain BDNF (brain-derived neurotrophic factor) level and activities of AChE (acetylcholinesterase) and antioxidant enzymes, and reduced malondialdehyde (MDA) level and caspase-3 activity. Since they found a positive correlation between serum 25(OH)D level and brain BDNF or AChE activity and a negative correlation between serum vitamin D level and brain MDA levels or caspase-3 activity, they concluded that increasing brain BDNF could be a key mechanism of vitamin D’s anti-aging effects, at least in the rat brain [94]. The progression of the brain aging process, which is often associated with dysfunctional hippocampal neurogenesis, has been hypothesized to be accelerated by vitamin D deficiency [95]. In such a scenario, vitamin D insufficiency in the senescent brain might cause changes in the proper function of Wnt signaling, which subsequently could lead to the failure in the control of the neurogenic homeostatic mechanisms involved in the protection of neural stem cells and, in this way, could evoke cognitive impairments. A better understanding of the functional link between vitamin D, neurogenesis and cognitive performance during aging will create space for designing new therapies for age-related cognitive decline [87].

## 3. Alzheimer’s Disease

Age-related dementia, including AD, is a major public health concern associated with growing healthcare costs. AD is a progressive ND which has major pathological features in the brain, including amyloid-β (Aβ) extracellular accumulation in the form of plaques and neurofibrillary tangle intracellular deposition. In addition to Aβ accumulation, a decline in its clearance can contribute to the progression of this disease. There is ample experimental evidence that vitamin D, due to its antioxidant, anti-inflammatory and neuroprotective properties, decreases Aβ neurotoxicity and facilitates Aβ enzymatic degradation and clearance via enhancement of LRP-1 (low-density lipoprotein-receptor-related protein) expression. Furthermore, 1α-hydroxylase, an enzyme responsible for the production of calcitriol, is present in the brain; the ability of this secosteroid to modify the production and release of neurotrophic factors supports the assumption of its potential neuroprotective function. It is assumed that vitamin D deficiency may increase AD risk and that vitamin D supplementation may be helpful in ameliorating neurodegeneration in AD patients, as evidenced by experimental in vitro (Table 1), in vivo (Table 2) and clinical studies.

### 3.1. Experimental Studies

There are many experimental studies showing the potency of vitamin D in the reduction of Aβ toxicity or pathological Aβ aggregation. Treating human amyloid precursor protein-expressing transgenic mice with calcitriol-lowered soluble Aβ levels, enhanced brain P-glycoprotein levels and reduced soluble and insoluble Aβ plaques, with the highest efficiency found in the hippocampus. These vitamin-D-evoked biochemical and morphological changes were accompanied with improved conditioned fear memory [96]. In an in vitro study, Pierucci et al. (2017) [97] found that calcitriol and its structural analog ZK191784 prevented Aβ(1–42) cytotoxicity in human neuronal-like model (SH-SY5Y neuroblastoma cells). Moreover, both compounds counteracted Aβ(1–42) peptide-induced cell damage through the prevention of the reduction in the sphingolipid S1P/S1P1 pathway induced by Aβ and consequent inhibition of p38MAPK/ATF4 signaling pathway. A further in vivo study performed by the same research group demonstrated that the Aβ(1–42)-induced damage in the CA1 region of the rat hippocampus was protected by chronic treatment with calcitriol, and this secosteroid in adult mice promoted cell proliferation in the hippocampal dentate gyrus [97]. In a recent study of the intracellular mechanisms of calcitriol in ameliorating Aβ toxicity in SH-SY5Y cells, it was found that this compound modulated Aβ-induced tau protein hyperphosphorylation, ROS and apoptosis. Furthermore, calcitriol increased VDR protein expression, restored the decreased GDNF (glia-derived neurotrophic factor) and the inhibited PI3-K/Akt/GSK3β (phosphatidylinositol 3 kinase/protein kinase B/glycogen synthase kinase-3β) intracellular pathway. Since the GDNF inhibitor heparinase III abolished the neuroprotective effects of calcitriol, it has been proposed that vitamin D alleviates Aβ neurotoxicity through interplay with GDNF signaling [98]. In addition to GDNF, the inducing effect of vitamin D on other neurotrophic factors can also contribute to its neuroprotective mechanism of action. Vitamin D3 and curcumin demonstrated similar protective effects against Aβ(1–42)-induced cell damage of primary cortical neuronal cultures, which was associated with attenuation of lipid peroxidation and increase in antioxidant enzymes (GSH and GST) and NGF levels [99]. Bao et al. (2020) [100] observed that calcitriol alleviated cognitive deficits in an AD rat model which was connected with modulation of the VDR/ERK1/2 signaling pathway. Moreover, this secosteroid in AD mice protected hippocampal neurons, inhibited apoptosis and increased p-ERK1 protein expression. Yu et al. (2011) investigated the effect of vitamin D supplementation on amyloid plaque formation in Aβ protein precursor (AβPP) transgenic mice, which, within 3–4 months of birth, spontaneously developed amyloid plaques. They found that AβPP mice fed with a cholecalciferol-enriched diet had lower number of amyloid plaques, decreased Aβ level and inflammation and increased NGF in the brains when compared to animals fed with a normal diet. Thus, it was concluded that vitamin D supplementation could be beneficial for AD patients [101]. In AD-transgenic (APPSwe/PS1dE9) mice, supplementation with vitamin-D2-enriched mushrooms caused improvement in learning and memory, a reduction in amyloid plaque load and GFAP protein levels and an increase in anti-inflammatory interleukin-10 (IL-10) [102]. In an AD mouse model exposed to D-galactose-induced oxidative stress, cholecalciferol chronic treatment improved neuronal synapses and memory, reduced oxidative stress, increased Nrf2 and HO-1 levels, and reduced neuroinflammation (NF-κB, TNF-α, IL-1β) [103]. Precognitive and neuroprotective effects of vitamin D3 supplementation in pre- and post-treatment schedule were investigated by Yamini et al. (2018) in a sporadic AD rat model induced by intracerebroventricular injection of streptozotocin (ICV-STZ) [88]. They evidenced that cholecalciferol pretreatment has superior efficiency to the post-treatment schedule and manifested in improved spatial learning and memory, reduced brain oxidative stress, mitochondrial aberrations, neuroinflammation and improved cholinergic functions [104]. Cognitive dysfunctions and the brain histopathological changes evoked by LPS in rats were improved by the active vitamin D analog maxacalcitol [89]. These effects were associated with increased expression of Nrf2 and its downstream effectors (HO-1 and GSH), improved serum 25(OH)D and calcium levels, reduced neuroinflammation, decreased Aβ load and tau hyperphosphorylation and inhibition of the ERK1/2 and p38 intracellular pathways. It should be added that Nrf2, the main transcriptional regulator of antioxidant response element (ARE), is regarded as a potential target for the prevention of AD [105]. Another process which has been implicated in AD is ferroptosis, a non-apoptotic form of cell death [19], which has been shown to be inhibited in transgenic APP/PS1 mice by paricalcitol. This VDR-specific ligand inhibited the brain Tau phosphorylation and decreased the iron-induced GSK3β activity [106]. Another evidence for the potential link between VDR and ferroptosis inhibition was provided by Li et al. (2022) [107], who demonstrated that flavone eriodictyol ameliorated cognitive deficit in APP/PS1 mice by VDR-mediated Nrf2 activation and inhibition of iron-dependent cell death.

There are many reports showing VDR expression in immune cells. and these genomic mechanisms could be involved in the immunomodulatory effects of vitamin-D-mediated neuroprotection [108]. It was found that calcitriol acting through the VDR potently induced the expression of the cytokine IL-34 in SH-SY5Y neuronal-like cells [93]. Since the cytokine IL-34 is associated with neuroprotective and survival signals in brain injury and neurodegeneration, it could be an immunological mediator for vitamin-D-induced protection from AD [109]. Another in vitro study showed that vitamin D attenuated the Aβ(25–35)-induced immune activation of BV-2 microglia cells, which was confirmed at the levels of ROS, IL-6, IL-1b, TNF-a, iNOS, COX-2 and NF-κB [110]. Vitamin D was also beneficial in AD rat models induced by intrahippocampal Aβ(1–40) injection, improving learning and memory and reducing neuronal loss, oxidative stress and neuroinflammation [111,112]. Grimm et al. (2017) investigated the effect of vitamin D (calcifediol) and its analogs used in clinical settings (maxacalcitol, calcipotriol, alfacalcidol, paricalcitol, doxercalciferol) on Aβ production and degradation. They found that vitamin D2 and vitamin D3 analogs, in a similar way as the parental compounds, decreased Aβ-production and increased Aβ-degradation in neuroblastoma SH-SY5Y cells or ex vivo in the vitamin-D-deficient mouse brain homogenates. These effects were manifested by a decrease in the activity of BACE1 and γ-secretase and the expression of nicastrin, as well as decreased β-secretase activity [113]. Transcriptomic analysis of the hippocampus and neocortex of WT and transgenic 5xFAD (Tg) mice after 5 months of cholecalciferol supplementation revealed a broad panel of dysregulated pathways. Among them, changes in immune and inflammatory responses, neurotransmitters, neurovascular unit and hormone levels were observed. As the authors noted, these differentially expressed genes were not directly dependent on VDR, but rather crosstalk with estrogen and insulin signaling in vitamin D action was suggested. It was concluded that the altered expression of the large number of genes found in this study could be responsible for improved learning and memory and a decrease in Aβ plaques and astrogliosis in Tg animals [114]. Using the same transgenic 5xFAD (Tg) mice, Kang et al. (2022) [115] investigated transcriptional and behavioral consequences of vitamin D depletion in early and late stages of development of AD symptoms in this model, as well as the effectiveness of cholecalciferol treatment. Vitamin D deficiency evoked in Tg mouse brains increased Aβ production and deposits, astrogliosis and memory impairments which were restored by cholecalciferol administration even at the late stage. Another study showed that calcitriol could reduce cerebral Aβ1–40 level by modulation of its BBB transport and peripheral uptake in the liver, which is connected with the VDR-dependent regulation of LRP-1 and RAGE (receptor for advanced glycation end products) protein expression [59]. Moreover, Patel et al. (2017), in their review paper, evidenced a strong correlation between vitamin D and LRP-1 expression and suggested their crucial role in Aβ clearance. They noted that among Aβ scavenger receptors, LRP-1 has been the most intensively investigated. Under physiological conditions, LRP-1 is highly expressed in neurons and located on the abluminal side of the brain capillaries; however, its expression is decreased in AD patients which could be the cause of increased cerebral Aβ depositions. Calcitriol treatment via VDR has been shown to increase LRP1 brain expression in various experimental studies and by this mechanisms reduce Aβ aggregates [116]. Other investigators found that paricalcitol, a low-calcemic vitamin D receptor agonist, reduced brain Aβ deposits and neurodegeneration in APP/PS1-transgenic mice. Paricalcitol stimulated LRP-1 expression, increased the lysosomal degradation of BACE1 (β-site APP-cleaving enzyme 1) and attenuated calpain-1-mediated neuronal loss in this animal model of AD [117]. Using a lipopolysaccharide (LPS)-based model of AD in rats, Medhat et al. (2020) found that vitamin D3 treatment or exercise markedly improved cognitive dysfunction and the histopathological picture of the rat AD brains, with the best results observed in the vitamin D + exercise group. These effects were associated with decreased IL-6, MDA, Aβ and tau proteins levels and increased IL-10, GSH, AChE, dopamine, BDNF and NGF. The authors concluded that combining vitamin D and exercise may be an effective method for treating AD [118]. Furthermore, in a well-established model of unpredictable chronic stress in rats, it was demonstrated that vitamin D3 elevated levels of BDNF, and this effect was accompanied by a decrease in Aβ content in the hippocampus. Thus, it was suggested that vitamin D treatment improves cognitive functions by increasing BDNF synthesis and inhibition Aβ production [119]. The cerebrovascular effects of vitamin D may also contribute to its protective effects against Aβ-induced neurodegeneration. As reviewed by Katusic and Austin (2014), there is emerging evidence that the loss of endothelial NO activates microglia and promotes a pro-inflammatory phenotype in the brain and may participate in the initiation and progression of cognitive decline, including the pathogenesis of AD. It has been shown that the loss of NO in cultured human cerebrovascular endothelium increases the expression of APP and BACE1 and the secretion of toxic Aβ1–40 and Aβ1–42. Moreover, enhanced expression of APP and BACE1 and production of Aβ were found in the cerebral microvasculature of eNOS-deficient mice [120]. Vitamin D supplementation can likely prevent endothelial dysfunction and the development of cerebrovascular disorders through its antioxidant and anti-inflammatory actions [121]. This notion is further supported by a study showing that vitamin D may prevent endothelial cell death by modulation of apoptosis and autophagy crosstalk, inhibition of superoxide anion generation, maintenance of mitochondrial function and cell viability, activation of pro-survival kinases, and induction of NO production [122]. It has been postulated that Aβ initiates neurodegenerative processes by robustly decreasing VDR expression. Deficiency of the translational coactivator of VDR, perioxisome proliferator-activated receptor-γ coactivator (PGC-1α), in transgenic mice resulted in decreased VDR expression and increased oxidative damage in the hippocampus and cortex. In contrast, overexpression of PGC-1α led to a significant increase in the VDR expression accompanied by a reduction in the Aβ plaque content in transgenic mice and of 8-oxo-dG in 2xTg-AD mice. These results suggest that the overexpression of PGC-1α could be a viable strategy for combating AD [123]. However, the prevailing evidence for the positive involvement of VDR in maintaining adult brain health has been recently challenged by the results of a study on AD postmortem human brains, APP/PS1 mice, and cell cultures [124]. The hippocampal tissue of AD patients showed elevated VDR expression, although the patients had decreased vitamin D serum levels. Moreover, the VDR colocalized with Aβ plaques, gliosis and autophagosomes, and Aβ upregulated VDR expression and changed its heterodimer binding partner from RXR to p53. The authors postulated that the conversion of genomic VDR/RXR into a nongenomic VDR/p53 signaling pathway might play a pivotal role in AD progression. This assumption was supported by a mechanistic study that showed that the VDR/p53 complex enhanced autophagy and apoptosis in neuronal cells, and that reinstatement of VDR/RXR by inhibiting p53 ameliorated amyloidosis and cognitive deficits in APP/PS1 mice [124]. The same team observed that APP/PS1 mice fed a vitamin-D-sufficient diet had markedly lower levels of serum vitamin D, suggesting its deficiency may be a consequence rather than a cause of AD. Furthermore, supplementation with vitamin D increased Aβ deposition and exacerbated cognitive impairments. They concluded that vitamin D supplementation in this animal model of AD did not reinstate the genomic VDR/RXR heterodimer but stimulated the formation of the non-genomic VDR/p53 complex in brain tissue [109]. However, the scientific rigor of this study has been questioned by Gombart et al. (2023) and Vieth (2022) [125,126]. In brief, the criticism of the Lai et al. (2022) [127] report is focused on insufficient in vitro evidence, very high non-physiological concentrations of vitamin D, a limited value of the AD model in mice, and inadequate discussion of the results of this study in the context of the prior literature.

**Table 1 cells-12-00660-t001:** Effects of vitamin D in experimental models of Alzheimer disease—in vitro studies.

AD Model	Vitamin D Administration	Effects	References
Aβ(1–42) peptide-induced toxicity in differentiated SH-SY5Y cells	calcitriol (100 nM)/24 h prior to Aβ(1−42) peptide (1 μM)/6 h	↑ cell viability ↑ sphingosine-1-phosphate, sphingosine kinase ↓ p38MAPK/ATF4 axis	[97]
Aβ(25–35) peptide induced toxicity in SH-SY5Y cells	calcitriol (0.1 or 10 nM)/24 h after Aβ(25–35) peptide	↑ cell viability ↑ VDR protein expression ↑ GDNF expression ↓ cell apoptosis ↓ ROS level ↓ p-tau/tau ↑ p-Akt/Akt	[98]
Aβ(1–42) peptide-induced toxicity in primary cortical neuronal cultures	vitamin D3 (1 nM), curcumin (5 μM), vitamin D3 + curcumin Aβ(1–42) (1 µM)/72 h	↓ lipid peroxidation ↑ reduced GSH ↑ GST enzyme level ↑ NGF	[99]
SH-SY5Y cells	calcitriol (10–500 nM)/48 h	↑ IL-34 mRNA and protein ↑ VDR expression	[109]
SH-SY5Y wt cells, SH-SY5Y APP695 cells and N2a cells	calcifediol, maxacalcitol, calcipotriol, alfacalcidol, paricalcitol, doxercalciferol (100 nM)/24 h	↓ Aβ-production ↑ Aβ-degradation ↓ γ-secretase activity ↓ β-secretase 1 ↓ IL-1β	[113]
Aβ(25–35)-induced damage of BV-2 microglia cells	ergocalciferol (1 μM)/1 h prior to Aβ(25–35) (20 μM)/24 h	↓ ROS, IL-6, IL-1β, TNF-α, iNOS, COX-2 and NF-κB activation	[110]
BBB model (bEnd.3 cells) HepG2	calcitriol (1–100 nM)/control or hypoxia/60 h/+ Aβ(1–40) (100 nM) calcitriol (1–100 nM)/6 h/ + Aβ(1–40) (115 nM)	↓ cerebral Aβ1–40 level ↑ LRP-1 expression ↑ VDR expression ↓ RAGE expression ↑ Aβ1–40 uptake	[59]
Hydrogen peroxide-induced damage of HUVEC line	vitamin D and VDR ligand (ZK191784) + hydrogen peroxide (200 μM)/20 min	↓ apoptosis-related gene expression, ↑ pro-autophagic Beclin 1 and the phosphorylation of ERK1/2 and Akt, ↑ mitochondrial potential ↓ cytochrome C release and caspase activation	[122]
NO-deprived BMECs	-	↑ APP, BACE1, Aβ(1–41) and Aβ(1–42) peptides	[120]

AD—Alzheimer’s Disease; Aβ—amyloid-beta peptide; Akt—protein kinase B; APP—amyloid precursor protein; ATF4—activating-transcription-factor-4; BACE1—amyloid precursor protein cleaving enzyme-1; BBB—Blood–Brain Barrier; bEnd.3—mouse brain microvascular endothelial cells line; BMECs—human brain microvascular endothelium; COX-2—cyclooxygenase-2; ERK—extracellular-signal-regulated kinase; GDNF—glial-cell-line-derived neurotrophic factor; GSH—gluthatione; GST—Glutathione S-transferase; HepG2—human hepatoblastoma cell line; HUVEC—human umbilical vein endothelial cells; IL—interleukin; iNOS—inducible nitric oxide synthase; LRP1—low-density lipoprotein-receptor-related protein 1; NGF—nerve growth factor; NF—κB-nuclear factor kappa-light-chain-enhancer of activated B cells; NO—nitric oxide; RAGE—receptor for advanced glycation end products; ROS—reactive oxygen species; tau—microtubule-associated protein; TNFα—tumor necrosis factor alpha; VDR—the vitamin D receptor.

**Table 2 cells-12-00660-t002:** Effects of vitamin D in experimental models of Alzheimer’s disease—in vivo studies.

AD Model	Vitamin D Administration	Effects	References
The human amyloid precursor protein-expressing models-Tg2576 and TgCRND8 mice	calcitriol (2.5 μg/kg, *i.p*., q2d × 4)	↑ brain P-glycoprotein ↓ soluble and insoluble plaque-associated Aβ ↑ conditioned fear memory	[96]
Intrahippocampal injection of Aβ(1−42) in rats Intraventricular injection of Aβ(1−42) in mice	calcitriol (1 μg/kg)/day/6 consecutive days before the hippocampal delivery of aggregated Aβ(1−42) and at days 8,11,13,15,18 after Aβ(1−42) calcitriol (1 μg/kg)/day/5 days before Aβ(1−42) and at days 8 and 12 after Aβ(1−42)	↓ neuronal damage in CA1 ↑ neurogenesis in dentate gyrus	[97]
Rats at different ages (5, 12 and 24 months)	cholecalciferol (500 IU/kg, orally)/day/5 weeks	↑ brain BDNF level, AChE, antioxidant enzymes; ↓ malondialdehyde level, caspase-3 activity	[94]
APPswe/PS1E9 and C57BL/6J mice	calcitriol (100 ng/kg), *i.p.*/daily/6 weeks	↑ learning and memory abilities ↓ morphological defects in hippocampal neurons ↓ caspase-3, Bax ↑ Bcl-2, VDR, p-ERK1	[100]
AβPP-PS1-transgenic mice	cholecalciferol (control diet, 2.4 IU/g), (Surplus vitamin D diet ~12 IU/g) or (vitamin D deficient diet, 0 IU/g)	↓ the number of amyloid plaques, Aβ peptides, TNF-α ↑ NGF	[101]
Two-month-old wild-type (B6C3) and AD transgenic (APPSwe/PS1dE9) mice	Diet deficient in vitamin D2 or a diet supplemented with vitamin-D2-enriched mushrooms (VDM), containing 160.2 mg/kg (54 IU/kg) vitamin D2/7 months	↑ learning and memory ↓ amyloid plaque load and glial fibrillary acidic protein ↑ interleukin-10 in the brain of VDM-fed wild type and AD transgenic mice	[102]
Intracerebroventricular STZ injection in rats	STZ (1.5 mg/kg)/first and third day cholecalciferol (42 IU, oral gavage)/day/pre-treatment 7 days before STZ/21 days and post-treatment 24 h after STZ/21 days	↑ spatial learning and memory functions ↓ neuronal oxidative stress ↓ mitochondrial aberrations ↑ cholinergic functions ↓ hippocampal neuroinflammatory response ↓ neuronal death in cortex and hippocampus	[104]
LPS-induced hippocampal damage in rats	Maxacalcitol (1 μg/kg, *i.p*.)/twice a day/4 weeks LPS (0.8 mg/kg, *i.p*./once a week/3 weeks before maxacalcitol	↓ TNF-α, MDA, ↑ Nrf2 ↑ IL-10, HO-1, GSH ↓ Aβ, p-Tau ↓ MAPK-p38, ERK1/2 ↓ neuronal damage in CA1 ↑ cognitive functions	[105]
APP/PS1-transgenic mice	Paricalcitol (200 ng/kg)/once every two days/15 weeks	↓ the iron accumulation in the cortex and hippocampus ↓ Transferrin receptor (TFR) and iron-regulatory protein 2 (IRP2) expression ↓ phosphorylation of Tau at Ser396 and Thr181 sites via inhibiting the GSK3β phosphorylation (Tyr216)	[106]
APPswe/PS1E9-transgenic mice	eriodictyol (50 mg/kg, *i.p*.)/3 times per week	↑ Nrf2/HO-1 ↑ VDR expression ↓ ferroptosis	[107]
D-galactose-induced oxidative stress in mice	cholecalciferol (100 μg/kg *i.p.*)/3 times a week/4 weeks	↑ the neuronal synapse and memory ↓ oxidative stress ↑ NRF-2, HO-1 ↓ NF-κB, TNF-α, IL-1β	[103]
mouse brain homogenates from WT and vitamin D deficient C57BL/6 mice	calcifediol, maxacalcitol, calcipotriol, alfacalcidol, paricalcitol, doxercalciferol (100 nM)/24 h	↑ Aβ-degradation ↓ β-secretase 1	[113]
5XFAD transgenic mice (Transcriptomic analysis)	cholecalciferol (7500 IU/kg)/5 months	improved dysregulated pathways related to immune, inflammatory response, neurotransmitter activity, endothelial, vascular processes, hormonal alterations	[114]
Kun Ming mice	calciriol (2.5 μg/kg, *i.p*./day/14 days	↑ LRP-1 expression ↑ VDR expression	[59]
Aging model in C57BL/6 fed a normal or high-fat/high-cholesterol diet	calcitriol (2.5 μg/kg, *i.p*.)/day/week	↑ P-glycoprotein expression	[93]
APP/PS1-transgenic mice	Paricalcitol (200 ng/kg, *i.p*.)/once every two days/15 weeks	↓ Aβ generation and neuronal loss ↑ LRP-1 expression ↑ lysosomal degradation of BACE1 ↓ 8-hydroxyguanosine generation in neuronal mitochondria resulting in the attenuation of calpain-1-mediated neuronal loss	[117]
LPS-treated rats	LPS (0.8 mg/kg, *i.p*.)/once a week/3 weeks + vitamin D3 (1 μg/kg)/twice a day/4 weeks and/or exercise (30 min/once daily/4 weeks	↑ cognitive dysfunction ↓ IL-6, MDA, Aβ, tau proteins levels ↑ IL-10, GSH, AChE, dopamine, BDNF, NGF	[102]
UCMS rat model	Vitamin D3 (100, 1000, 10,000 IU/kg, *i.p*.)/4 weeks during UCMS	↓ corticosterone ↑ spatial learning and memory ↓ oxidative stress ↓ Aβ ↑ BDNF	[118]
eNOS-deficient mice	-	↑ APP, BACE1, Aβ peptides	[120]
APP/PS-1 double-transgenic (2xTg-AD) mice	-	↓ PGC-1α ↓ VDR	[123]
Intrahippocampal Aβ(1–40) peptide injection in rats	calcitriol (1 µg/mL×1 mL/kg, *i.p*.)/day/14 days	↑ learning and memory ↓ neuronal loss and oxidative stress (↓ MDA, ↑ SOD)	[111]
Intrahippocampal injection of Aβ(1–40) peptide in rats	cholecalciferol (5 μg/kg/day, *i.p*.)/2 weeks	↓ Aβ-induced memory impairment ↑ antioxidant markers (TAC, TTG) ↓ stress oxidative biomarkers (LPO, DNA damage)	[112]
Double-transgenic APP/PS1 mice	vitamin D3-sufficient diet (600 IU/Kg of cholecalciferol) vitamin D3-supplemented diet (8044 IU/Kg of cholecalciferol)	↓ serum level of vit. D3 ? ↑ Aβ deposition ? ↑ cognitive impairments ?	[127]
Transgenic hemizygous 5xFAD mice	vitamin-D-deficient diet cholecalciferol (410 ng/g, *i.p*.)	↑ Aβ production and deposits, astrogliosis, memory impairments- restored after cholecalciferol administration	[115]

AChE—acetylcholinesterase; AD—Alzheimer’s Disease; Aβ—amyloid-beta peptide; BACE1—amyloid precursor protein cleaving enzyme-1; Bcl-2—family of regulator proteins that regulate cell death (apoptosis); Bax—pro-apoptotic Bcl-2 family member; BDNF—brain-derived neurotrophic factor; eNOS—endothelial nitric oxide synthase; ERK—extracellular signal-regulated kinase; GSH—gluthatione; GSK3β—serine⁄threonine protein kinase; HO-1—heme oxygenase 1; IL—interleukin; LPO—lipid peroxidation; LPS—lipopolysaccharide; LRP1—Low-density lipoprotein receptor-related protein 1; MDA—malondialdehyde; NF—κB-nuclear factor kappa-light-chain-enhancer of activated B cells; NGF—nerve growth factor; Nrf-2—nuclear-factor-erythroid-2-related factor 2; PGC-1α—transcriptional coactivator that regulates the genes involved in energy metabolism; SOD—superoxide dismutase; TAC—total antioxidant capacity; tau—microtubule-associated protein; TNFα—tumor necrosis factor alpha; TTG—total thiol groups; STZ—streptozotocin; UCMS—unpredictable chronic mild stress; VDR—the vitamin D receptor; ?—questionable data.

### 3.2. Clinical Studies

A number of clinical studies have attempted to find a correlation between serum 25(OH)D (calcidiol) deficiency and cognitive impairments in various forms of dementia. The clinical practice guidelines of the Endocrine Society Task Force on Vitamin D classify serum vitamin D level as sufficient >30 ng/mL, insufficient 20–30 ng/mL, and deficient <20 ng/mL [128]. The calcidiol level is usually used to estimate systemic vitamin D status because it is more stable and is present at much higher concentrations in the blood than calcitriol [129,130]. 

In a study of 44 AD patients treated with AChE inhibitors (donepezil, rivastigmine or galantamine), 26 untreated AD patients and a control group (*n* = 35), it was found that the serum levels of 25(OH)D2 and 25(OH)D3 were the lowest in untreated AD patients and the highest in the treated AD group [131]. The authors stated that low levels of vitamin D in AD patients result from very low levels of both 25(OH)D2 and 25(OH)D3 and that this deficit can be reversed by treatment with AChE inhibitors. However, no relationship was found between cognitive function score and any form of vitamin D serum level [131]. In a further study, Shah et al. (2014) [132] used a liquid chromatography–tandem mass spectrometry (LC-MS/MS) assay to quantify 10 metabolites of vitamin D in sera from healthy adults and patients with AD, type 1 diabetes and rheumatoid arthritis. The results showed higher concentrations of the metabolite 3-epi-25(OH)D3 in the samples from patients with disease, stressing the importance of vitamin D epimers as useful disease-related biomarkers. As reviewed by Soni et al. (2012), associations have been found between low 25(OH)D and AD and dementia in both Europe and the US. This group noted up to four times higher risk of cognitive impairment in calcidiol-deficient elderly individuals (<25 nmol/L) than in individuals with adequate levels (≥75 nmol/L) [133]. Lee et al. (2009), in a multicenter clinical study, examined the link between serum 25(OH)D level and cognitive functions in middle-aged and older European men. They found that in a total of 3133 elderly men included in the analysis, the mean 25(OH)D concentration was 63 nmol/L, and a lower calcidiol level was associated with worse performance on the cognitive function test (Digit Symbol Substitution Test), with the best correlation between both factors found for 25(OH)D below 35 nmol/L [134]. Littlejohns et al. (2014) found a similar observation in a clinical study on 1658 elderly ambulatory adults free from dementia, cardiovascular disease, and stroke. They found that during a mean follow-up of 5.6 years, 171 participants developed dementia, including 102 cases of AD, and estimated that the risk for development of these diseases markedly increased when vitamin D concentration was below 50 nmol/L. Thus, an association between an increased risk of all-cause dementia and AD and vitamin D deficiency has been postulated [135]. More recently, as reviewed by Aspell et al. (2018), epidemiological evidence appears to support a link between low serum calcidiol level and worsened cognitive performance in community-dwelling older populations. However, due to lack of interventional evidence, an optimal 25(OH)D level to maintain good cognition is not established, and it remains unclear if increasing 25(OH)D concentrations will improve cognitive function. As the authors suggest, it seems justified to prevent vitamin D deficiency and use known common protective lifestyle factors in older adults to keep the brain healthy [136]. A meta-analysis of data from seven prospective cohort studies and one retrospective cohort study (total *n* = 28,354) involving 1953 cases of dementia and 1607 cases of AD revealed that higher levels of serum 25(OH)D were associated with a lower risk of dementia and AD, but no conclusive evidence regarding serum 25(OH)D levels of >35 ng/mL was obtained [137]. A meta-analysis of twelve prospective cohort studies and four cross-sectional studies showed significant correlation between vitamin D deficiency and both dementia and AD. Notably, there were stronger links between severe vitamin D deficiency (<10 ng/mL) and both dementia and AD than moderate vitamin D deficiency (10–20 ng/mL) [138]. However, some clinical studies do not support the hypothesis that vitamin D insufficiency is associated with AD pathology and that this vitamin is important for maintaining cognitive function in elderly individuals. The multicenter European Male Aging Study (EMAS) of 3369 men aged 40–79 showed that 25(OH)D and 1,25(OH)2D levels were associated with specified measures of cognitive decline in aging men; however, they disappeared after adjusting the data for additional factors such as depressive symptoms, BMI and comorbidities. Since no relationship between 1,25(OH)2D and cognitive decline was found, it was concluded that there was no evidence for an independent association between 25(OH)D or 1,25(OH)2D levels and visuoconstructional abilities, visual memory or processing speed over an average of 4.4 years in this sample of middle-aged and elderly European men [139]. Olsson et al. (2017) evaluated the relationship between vitamin D and the risk of dementia in a cohort of 1182 Swedish men (mean age 71 y) in a maximum of 18 years of follow-up. Since plasma calcidiol, vitamin D intake from the diet, and vitamin-D-synthesis genetic risk score (GRS) were not associated with any cognitive outcomes, it was concluded that there was no association between baseline vitamin D status and long-term risk of dementia or cognitive impairment [140]. A meta-analysis of prospective cohort studies with data on serum vitamin D level and AD risk involving six prospective cohort studies with 1607 AD cases and 21,692 individuals indicated that serum vitamin D deficiency (<25 nmol/L) or insufficiency (25–50 nmol/L) was not correlated with AD risk. Thus, the results from longitudinal studies investigating the association of vitamin D with dementia and cognitive impairment are inconsistent [141]. Panza et al. (2021) analyzed recent evidence for the effects of vitamin D insufficiency on AD and the role of supplementation and found that an increased susceptibility to AD could be the effect of insufficient (25–49.9 ng/mL) and deficient of vitamin D levels (<25 ng/mL). They pointed to the need for further prospective studies to better understand the involvement of low vitamin D status in the pathomechanism of AD and efficacy of vitamin D supplementation In its prevention or treatment [142]. In contrast to the large number of studies on circulating vitamin D, less is known about vitamin D level in the human brain. A recent report confirmed that a reasonable level of calcidiol in human brain structures is engaged in cognitive functions. Higher brain 25(OH)D3 level was associated with better cognitive function prior to death; however, it was not associated with neuropathological changes [143].

The pivotal question Is whether treatment with vitamin D3 may ameliorate or slow cognitive deficits in progressive dementia disorders. As noted by Annweiler (2016), although no randomized placebo-controlled trials have been conducted to study the effectiveness of vitamin D administration to prevent AD, various nonrandomized controlled studies have found that older adults experienced cognitive improvements after 1–15 months of vitamin D supplementation. It has also been suggested that the combination of memantine, the only drug that shows moderate clinical efficacy in ameliorating cognitive decline in the early stages of AD, with vitamin D may represent a new multitarget neuroprotective approach in AD and related disorders (ADRD). The AD-IDEA trial (a unicenter, double-blind, randomized, placebo-controlled, intent-to-treat superiority trial) compared 24 weeks of oral intake of vitamin D3 (cholecalciferol) treatment with the effect of a placebo on the change in cognitive performance in one hundred and twenty participants with moderate ADRD receiving memantine, but the results of this trial have yet to be reported [144]. Clinical studies have also demonstrated new data on biochemical mechanisms of vitamin D in AD. Lu’o’ng et al. (2011) reviewed experimental and clinical evidence for the putative role of vitamin D in AD pathology and pointed out that patients with AD have a high prevalence of vitamin D deficiency. Moreover, they postulated that vitamin D has a beneficial role and improves cognitive function in some patients with AD. On the other hand, the engagement of multifunctional biochemical factors and signaling in the mechanism of vitamin D action has been underlined. Genomic mechanisms by which vitamin D via the VDR can affect AD pathology include the modulation of multiple protein expression levels (the major histocompatibility complex class II molecules, renin-angiotensin system, apolipoprotein E, liver X receptor, Sp1 promoter gene, and the poly(ADP-ribose) polymerase-1 gene). Furthermore, non-genomic effects of vitamin D3, including regulation of L-type voltage-sensitive calcium channels, the prostaglandins, NGF, cyclooxygenase 2, ROS and nitric oxide synthase, were also found to affect disease progression [145]. Since vitamin D upregulates insulin receptor expression and nasal insulin improves cognition, Stein et al. (2011) [146] evaluated the impact of high-dose vitamin D followed by intranasal insulin administration on memory and disability in 32 mild-moderate AD patients in a randomized controlled trial. This group found that high-dose vitamin D did not increase the beneficial effect of 8 weeks of treatment with the low dose of vitamin D on memory or disability in mild-moderate AD patients. In an other randomized controlled trial, 12 months of vitamin D administration (800 IU/day) decreased biomarkers linked with Aβ and improved memory in elderly AD patients [147].

In their recent review, Marazziti et al. (2021) postulated that vitamin D likely plays a key role in the inflammatory response that is currently hypothesized to be involved in the pathophysiology of various psychiatric disorders. Vitamin D deficiency is associated with the pro-inflammatory state and the formation of Aβ oligomers that might contribute to the cognitive deficits observed with increased age. This group suggested that “vitamin D supplementation might pave the way towards “natural” treatments of a broad range of neuropsychiatric disorders, or at least be useful to boost response to psychotropic drugs in resistant cases” [148]. SanMartin et al. (2018) studied whether the correction of low vitamin D levels protected lymphocytes from oxidative death and elevations of Aβ1–40 plasma levels in mild cognitive impairment (MCI) and very early AD (VEAD) patients. In MCI patients, but not VEAD patients, lymphocyte susceptibility to death, as well as Aβ1–40 plasma levels, improved after 6 months of vitamin D treatment. Additionally, cognitive functions at follow-up (18 months) were revised in MCI patients after vitamin D supplementation. It was concluded that vitamin D supplementation might be favorable in MCI, but not in the more advanced stage of the neurodegenerative process [149]. In contrast, Lai et al. (2022) [127] concluded, based on a retrospective, population-based longitudinal study, that vitamin D3 supplementation increased the probability of developing dementia in older adults and increased the risk of mortality in patients with pre-existing dementia. However, an interpretation of these data appears to be quite inaccurate because the participants were treated with calcitriol, not as a dietary supplement, but as a strong medication prescribed to dementia-predisposed patients. It was strongly emphasized that the study by Lai et al. (2022) had serious methodological and interpretative limitations, confusing a drug commonly used for patients with kidney failure with the nutritional use of vitamin D, and, therefore, does not justify the conclusions that “vitamin D supplementation worsens Alzheimer’s progression” [125,126]. A potential vasculo-protective role of vitamin D in dementia patients was suggested by Buell et al. (2010) [150]. The obtained results showed that vitamin D insufficiency and deficiency were associated with all-cause dementia, AD and stroke (with and without dementia symptoms), as well as MRI indicators of cerebrovascular disease. A longitudinal cohort study in 10,186 participants from Danish general populations revealed an association of reduced plasma 25(OH)D with enhanced risk of the combined end-point of AD and vascular dementia [151]. Thus, the importance of the vascular effects of vitamin D in the pathomechanisms of AD and other forms of dementia should be stressed.

It has been suggested that SNPs in the *VDR* gene, which alters the affinity of vitamin D3 to its receptor, can contribute to the pathomechanism. Gezen-Ak et al. (2007) carried out a clinical trial including 104 AD patients and 109 age-matched controls, who were genotyped to detect differences in restriction and no restriction sites in intron 8 and exon 9 of the ligand-binding site of the *VDR* gene. This study found a significant correlation between the *VDR* gene and late-onset AD, indicating a protective role of the “AT” haplotype in AD and over two-times-higher risk of developing AD in the “Aa” genotype when compared with the “AA” genotype [152]. The associations of *VDR* gene SNPs with AD have been investigated. Liu et al. (2021) [153] conducted a meta-analysis of reports published before 30 October 2020 to verify the possible involvement of *VDR* gene polymorphisms to AD and MCI susceptibility, including ten case–control studies with 3573 participants and four loci: ApaI rs7975232, BsmI rs1544410, FokI rs10735810 and TaqI rs731236. This analysis revealed that depending on the population, *VDR* ApaI, BsmI and TaqI gene polymorphisms might be predictors of AD or MCI susceptibility. Another meta-analysis suggested that the association of the *VDR* gene polymorphism (ApaI) and the occurrence of AD might partly depend on ethnic origin as well as climatic conditions, since an association was found in populations of the UK and Poland but not Iranian and Turkish populations [154]. Very recently, Dimitrakis et al. (2022) [155] reported that in the Southeastern European Caucasian population, TAC (TaqI, BsmI and FokI) and TA (TaqI and BsmI) are risk haplotypes for late-onset AD, while the CAC (TaqI, BsmI and FokI) haplotype may act protectively. Other investigators using full-length 16S ribosomal RNA (rRNA) gene sequencing found that the crosstalk between gut microflora and vitamin D receptor variants is associated with the risk of amnestic MCI in the elderly Chinese population [156]. It has been found that patients with positive biomarkers of AD have low 25(OH)D levels in cerebrospinal fluid, but not in the serum, and that CSF levels of 25(OH)D were not related to any of the polymorphisms of the *VDR* gene: FokI, BsmI, ApaI and TaqI (Soares et al. (2022)) [157]. They also suggested that those with TaqI and BsmI major homozygote genotypes might be at increased risk for the development of dementia. In a Chilean cohort study, alleles A of Apa I and C of Taq I were associated with a lower risk of MCI, and the authors indicated that in carriers of protective alleles of Apa I polymorphism, the response to vitamin D treatment could be more efficient [158]. On the other hand, de Oliveira et al. (2018) [159] found no association among the four polymorphisms on the *VDR* gene (BsmI, ApaI, FokI and TaqI), cognitive deficits and reduction in vitamin D levels in patients with AD and MCI; however, BsmI polymorphism was associated with vitamin D levels in individuals with cognitive decline.

## 4. Parkinson’s Disease

PD is the second most prevalent age-related ND, and its clinical symptoms comprise bradykinesia, resting tremor, abnormal posture balance and hypermyotonia. The hallmarks of PD are the aggregation of misfolded α-synuclein in intracellular inclusions, referred to as Lewy bodies, and the degeneration of dopaminergic neurons in the nigrostriatal pathway, although other neuronal systems can also be affected [38,160]. Therefore, in experimental models of this disease, some neurotoxins of dopaminergic neurons, e.g., 6-hydroxydopamine (6-OHDA), 1-methyl-4-phenylpyridium ion (MPP(+)) and rotenone, are widely employed [161]. Vitamin D regulates numerous genes in the CNS by binding to the VDR; the VDR and 1α-hydroxylase, the enzyme that converts vitamin D to its active form, are highly expressed in the substantia nigra. A hypothesis has been formulated that vitamin D deficiency is associated with the pathomechanisms of NDs, including PD. Indeed, a high prevalence of vitamin D deficiency has been detected in PD patients. Some experimental data support the abovementioned hypothesis (Table 3 and Table 4), as do some clinical trial data. 

### 4.1. Experimental Studies

It has been reported that calcitriol ameliorates cell damage in rat mesencephalic culture induced by a 24 h exposure to 6-OHDA or MPP(+) by facilitating cellular functions that reduce oxidative stress [162,163]. Furthermore, low doses of calcitriol reduced damage of dopaminergic neurons in rat mesencephalic culture exposed to a mixture of L-buthionine sulfoximine (BSO) and MPP(+) and attenuated the generation of ROS and glutathione (GSH) depletion in this in vitro model [164]. The protective effect of calcitriol on levodopa (L-DOPA)-induced neural stem cell (NSC) injury has been reported. Exposure of the cells to L-DOPA increased the levels of free radicals and reduced cell viability and proliferation. Calcitriol protected NSCs against L-DOPA-induced injury by promoting prosurvival signaling, including activation of the PI3-K pathway, and reducing oxidative stress [165]. The same investigators reported that this active vitamin D3 metabolite protected SH-SY5Y cells against rotenone-induced damage by enhancing autophagy signaling pathways that involved LC3 and Beclin-1, providing an experimental basis for its clinical use in the treatment of PD [166]. During development, the VDR is expressed in midbrain tissue and can engage in signaling mechanisms that control differentiation, maturation and survival of developing neurons. Calcitriol, when added to primary fetal ventral mesencephalic cultures, upregulated the expression of GDNF and increased the number of dopaminergic neurons. Moreover, in apoptosis assays and cell birth dating experiments, calcitriol increases the number of dopamine neurons through neuroprotection but does not stimulate their differentiation [167]. In the model of α-Syn-induced aggregation in SH-SY5Y cells, a relatively high concentration of vitamin D (4 μM) improved cell viability, reduced cytotoxicity of α-Syn oligomers and decreased ROS [168].

Regarding in vivo studies, it has been reported that 6-OHDA-induced neurotoxicity in rats can be attenuated by vitamin D3 treatment, which was associated with an improvement in locomotor activity and an increase in dopamine and its metabolites (DOPAC and HVA) level in the substantia nigra (SN) [163]. Cass and Peters (2017) [169] investigated the beneficial effect of calcitriol against 6-OHDA-induced lesions in the striatum depending on rat age and showed that although this secosteroid increased the levels of DA in young, adult and aged animals, at the striatal level, it increased the evoked overflow of DA from the lesioned striatum only in young adult and middle-aged rats. Thus, they concluded that the effectiveness of calcitriol could be reduced in aging. In the rat model of PD induced by 6-OHDA striatal lesion, it was found that pre- or post-treatments with vitamin D3 reversed behavioral changes and improved the decreased DA contents of the 6-OHDA group. Furthermore, vitamin D3 reduced oxidative stress, increased TH and DAT and reduced TNF-α immunostainings in the lesioned striata. While significant decreases in VDR immunoreactivity were observed after 6-OHDA lesioning, these changes were blocked after vitamin D3 pre- or post-treatments. This study showed that vitamin D3 offers neuroprotection, decreasing behavioral changes, DA depletion and oxidative stress. In addition, vitamin D3 partially or completely reversed the TH, DAT, TNF-α and VDR decreases in immunoreactivities in the untreated 6-OHDA group. It was concluded that vitamin D3 effects could result from its anti-inflammatory and antioxidant actions by VDR genomic action [170]. Some data indicate that vitamin D3 may ameliorate the endothelial vascular impairment that accompanies neurodegeneration in PD. In this regard, Kim et al. (2020) reported that 1,25(OH)2D3 reversed the reduction in the transcription of the *VDR* and its downstream target genes *CYP24* and *MDR1a* and the vascular protein expression of P-gp in a 6-OHDA-induced PD mouse model. Furthermore, this group found that the 6-OHDA-induced decrease in endothelial P-gp protein was also reversed by 1,25(OH)2D3 [171]. Another study employing a 6-OHDA PD model in mice demonstrated that treatment with cholecalciferol improved behavioral deficits, reduce oxidative stress, increased level of the dopaminergic markers (TH, DAT), stimulated BDNF, and reduced MAO-B activity and neuroinflammation [172]. Recently, vitamin D has been shown to intensify the effects of exercise in a hemiparkinsonian rat model (6-OHDA-based) by improving behavioral parameters and reducing brain oxidative stress and neuroinflammation [173].

**Table 4 cells-12-00660-t004:** Effects of vitamin D in experimental models of Parkinson’s disease—in vivo studies.

PD Model	Vitamin D Administration	Effects	References
6-OHDA-induced neurotoxicity in rat	calcitriol (1 μg/mL × 1 mL/kg, *i.p.* per day)/8 consecutive days	↑ locomotor activity ↑ DA, DOPAC, HVA	[163]
6-OHDA-induced lesion in rat	calcitriol (1 μg/kg)/day by gavage/7 days before 6-OHDA-induced lesion calcitriol (1 μg/kg)/day by gavage/14 days after 6-OHDA-induced lesion	↓ behavioral changes ↓ DA depletion ↑ neuroprotective effects ↑ oxygen consumption rate ↓ mitochondria swelling, H_2_O_2_ production, SOD activity ↑ TH, DAT, VDAC, Hsp60 expression	[170]
6-OHDA-induced lesion in mouse	calcitriol (2.56 μg/kg, *i.p*.)/once every two days (total 4 times)	↓ dopaminergic loss ↓ neuroinflammation ↑ transcriptional VDR, VDR target genes ↑ endothelial P-glycoprotein	[171]
6-OHDA -induced lesion in young adult (4 month old), middle-aged (14 month old) and aged (22 month old) rat	calcitriol (1.0 μg/kg, s.c.)/4 weeks after 6-OHDA (12 μg)/once a day/8 consecutive days	↑ overflow of DA from the lesioned striatum, ↑ striatal DA in the young adult and middle-aged rats ↑ DA in substantia nigra in all age groups	[169]
6-OHDA-induced lesion in mouse	cholecalciferol and/or L-DOPA/2 weeks post lesion	↓ behavioral deficits, ↓ protein associated with dopamine metabolism ↓ biomarkers of oxidative stress ↑ contralateral wall touches, exploratory motor and cognitive activities ↑ TH, DAT, BDNF expression of, while ↓ MAO-B, CD11b, IL-1β and p47phox expression	[172]
6-OHDA-induced lesion in rat	cholecalciferol (1 µg/kg)/21 days, in the absence and presence of physical exercise on a treadmill (30 min, speed of 20 cm/s, once a day/21 days/24 h after 6-OHDA	↑ behavioral changes ↑ DA, DOPAC, TH, DAT ↑ VDR ↓ oxidative stress ↓ nitrite	[173]

BDNF—brain-derived neurotrophic factor; DAT—dopamine transporter; DOPAC—3,4-dihydroxyphenylacetic acid; HVA—homovanillic acid; MAO-B—monoamine oxidase B; 6-OHDA—6-hydroxydopamine; p47phox—component of the NADPH oxidase; PD—Parkinson’s disease; SOD—superoxide dismutase; TH—tyrosine hydroxylase; VDAC—voltage-dependent anion-selective channel protein; VDR—the vitamin D receptor.

### 4.2. Clinical Studies

Luong and Nguyên (2012) [174] pointed out that the role of vitamin D in PD might be favorable. These authors underlined a significant association between low concentrations of vitamin D in the serum and PD, and suggested that vitamin D supplementation appears to have a positive clinical effect on PD patients. Of importance are genetic data that point to VDR-regulated genes, which can be involved in the pathomechanism of PD, e.g., Nurr1 gene, Toll-like receptor and tyrosine hydroxylase. It is postulated that calcitriol is most appropriate for the treatment of PD because it modulates the expression of inflammatory cytokines, implicated in the neuroinflammatory mechanisms of PD. The risk of developing PD may be due to sun exposure as well as vitamin D concentration in earlier life. Data of cross-sectional research suggest an interaction between serum vitamin D concentrations and the PD symptoms. However, although malnutrition status was linked with worse clinical severity and more severe cognitive deficits, a low vitamin D level alone did not correlate with such symptoms in PD patients [175]. One intervention study found some beneficial effects of vitamin D administration in PD. The authors concluded that although vitamin D3 may affect PD symptoms and the risk of the appearance or progression of PD, more studies are needed to confirm this hypothesis [176]. Wang et al. (2015) [177] measured plasma total 25(OH)D, 25(OH)D2, and 25(OH)D3 levels in 478 PD patients (*n* = 478) and 431 controls. The results demonstrated a negative association between 25(OH)D2 deficiency/insufficiency and PD risk. The authors concluded that the dependence between vitamin D concentration and PD could be linked not only to restricted sunlight exposure but also to gastrointestinal dysfunction in PD patients. The relationship between serum vitamin D and neuropsychiatric function was studied in 286 PD patients. Significant associations were found between vitamin D concentration and verbal fluency and verbal memory, with better mood estimated in depression scores in PD participants not suffering from dementia [178]. The biological correlations between vitamins and PD and an elaboration on the therapeutic potentials of vitamins for PD was summarized by Zhao et al. (2019). The authors suggest that the antioxidant activity of vitamins and their potential in regulating gene expression may be crucial for the treatment of PD, while the supplementation of vitamins, including vitamin D3, can diminish the number of PD patients in the general population and improve the clinical symptoms. It is proposed that vitamin supplementation may be an effective adjuvant treatment for PD [179]. Very recently, the correlations between vitamin D and PD symptoms have been comprehensively summarized by Fullard et al. (2020) [180]. Inconsistencies in the reported data were noted, as one prospective study found an elevated risk for PD with lower mid-life vitamin D levels, while another pointed to the lack of association between PD risk and vitamin D levels. However, more consistent data have been demonstrated in cross-sectional studies regarding the inverse correlation motor symptom severity and between serum vitamin D levels. It is postulated that further research on vitamin D assessment in the course of PD, as well as the treatment for those with vitamin D deficiency, is required [180]. In a study that included 182 PD patients and 185 healthy controls, vitamin D serum concentrations correlated well with falls, insomnia and scores for sleep quality, anxiety and depression, but no with bone mineral density in PD patients [181].

The dependence between cerebral vascular disease and PD remains a matter of debate; however, several types of simultaneous occurrences of parkinsonism and cerebral vascular disease are currently recognized. The postulated concept of vascular parkinsonism is controversial, but in elderly individuals, cerebrovascular diseases and idiopathic PD may coincide [182,183]. Nevertheless, vitamin D status is a known biological factor that modulates vascular endothelial function with aging; thus, its deficiency might contribute to the age-dependent development of PD [184]. 

Studies on the possible association of *VDR* gene polymorphisms and the exposure and severity, as well as the cognitive deficits, of PD have provided inconsistent results [129]. Butler et al. (2011) [185] tested 18 VDR SNPs and found that common VDR genetic variations modulate the age-at-onset of PD. Moreover, Lv et al. (2020) [186] reviewed studies on the associations among vitamin D serum concentrations, VDR gene polymorphisms and PD, concluding that although data regarding gene polymorphisms and the risk of PD are inconsistent, the FokI (C/T) gene polymorphism may increase PD risk, severity and memory deficit. Similarly, the results of genetic studies examining polymorphisms of the VDR and PD risk, severity or age at onset conducted by Peterson (2014) [176] have provided variable results, with FokI CC possibly associated with increased risk for PD. Among the four best recognized VDR SNPs (rs731236 (TaqI); rs7975232 (ApaI); rs1544410 (BsmI); and rs2228570 (FokI), the Fokl and Apal are likely associated with PD, and might influence the vitamin D3 supplementation effect in patients [187,188]. Agliardi et al. (2021) [189] genotyped ApaI, BsmI, TaqI, FokI and rs1989969 VDR SNPs in a cohort of 406 patients suffering from PD and 800 healthy controls and demonstrated a strong association between the FokI (rs2228570) VDR SNP and parkinsonism. The FokI SNP was also implicated in an elevated exposure to sporadic PD in a study population in China covering 940 subjects [187]. In the Hungarian population, a strong relationship between the VDR FokI CC genotype and milder forms of PD has been observed [190]. The Fokl CC genotype was also more frequent in Japanese PD patients with higher concentrations of vitamin D and milder motor severity [191]. Furthermore, it was found that VDR SNP rs2228570 may be related to sporadic PD in Japan, while smoking did not affect this relationship [192]. Interestingly, a case–control study conducted in 190 Caucasian patients with PD found that FokI was associated with cognitive impairment [193]. Moreover, the data from the same research group found associations between TaqI and ApaI, but not FokI, loci and the risk of PD, in a population with high ultraviolet radiation exposure [194]. Recently, Redenšek group (2022), in a study on a Slovenian cohort of 231 patients suffering from PD confirmed the association between VDR rs2228570 and the risk of the disease development. Moreover, they reported strong associations between VDR polymorphisms and non-motor adverse events related to dopaminergic treatment, including visual hallucinations and orthostatic hypotension. As the authors concluded, their data support a personalized approach to treatment of PD [195]. Gao et al. (2020) [196] conducted a meta-analysis of data on the association between vitamin D receptor polymorphisms and susceptibility to PD, including 3194 controls and 2782 PD patients in Asian populations, and found the Fokl gene polymorphism to be the only risk factor for PD.

## 5. Conclusions

The biological effects of vitamin D are mainly exerted by its nuclear receptor. In addition to regulating calcium and phosphorus homeostasis, vitamin D has profound effects on neuroinflammatory and neurodegenerative processes within the CNS via both genomic and nongenomic mechanisms. The mechanisms of the neuroprotective effects of the active form of vitamin D are very complex and include both VDR-mediated and nongenomic effects on neurotrophin expression, downregulation of L-type calcium channel expression, attenuation of oxidative stress, excitotoxicity, apoptosis and ferroptosis, and promotion of antiaging/neuroprotective processes through interference with multiple prosurvival intracellular signaling pathways (Figure 2). The ability of vitamin D to mitigate neuroinflammation is of great importance, as reflected by the decreased expression and release of some proinflammatory cytokines and nitric oxide. In a recent systematic review, Iacopetta et al. (2020) attempted to estimate whether the route of administration of vitamin D has an impact on of its neuroprotective benefits in NDs by comparing the efficiency of endogenously produced vitamin D from sun exposure to exogenous synthetic supplementation of vitamin D. This group found no solid evidence to support the hypothesis that natural synthesis of vitamin D is responsible for the beneficial effect of UV exposure in Multiple Sclerosis (MS), PD and AD. It was concluded that it is rather uncertain if vitamin D has a protective benefit in NDs or whether it is associated with UV exposure, which may mediate currently unidentified neuroprotective factors [197]. Bivona et al. (2019) discussed low vitamin D serum levels found in patients with AD, PD, MS, Autism Spectrum Disorders, Sleep Disorders and Schizophrenia; however, in order to better understand the efficiency of vitamin D supplementation in these brain disorders, further rigorous clinical trials are needed [198]. Additionally, Morello et al. (2020), in their review, underlined that only further research on vitamin D will help to understand how this secosteroid can help to optimize the prevention and treatment of several neurological diseases [199]. The conflicting current clinical data may result from inadequate dosing, high variability in the designing of the trials which hinders comparative analysis of their results, various routes of administrations used and insufficient consideration of the age- and sex-dependency of vitamin D3 pharmacological effects. The most effective vitamin D blood level for slowing the progression of age-related detrimental processes within the CNS remains to be determined. An encouraging notion is that the results of most studies, despite the aforementioned limitations and shortcomings, suggest that supplementation with vitamin D3 can be beneficial in preventing or attenuating age-related brain disorders, at least in some patient subpopulations. Association studies on relationships between functional VDR polymorphisms and susceptibility to NDs may help complement personalized approaches to treating these disorders. In addition to the direct neuroprotective and anti-inflammatory properties of vitamin D3, its influence on the neurovascular system and cerebral blood flow should be considered when developing new neuroprotective strategies. It should be mentioned that there is still a great interest in designing novel VDR ligands. For example, novel des-D-ring interphenylene analogs of vitamin D3 have been recently designed, and some of them have shown high gene-transcription-promoting activities comparable to natural 1α,25(OH)2D3. In particular, the analogs with a meta-substituted side chain display high potency to form the complex with VDR and co-activator [200]. However, biological properties of these analogs have not been described yet. Regarding the therapeutic potential of these compounds in the treatment of neurodegenerative diseases, non-calcemic analogs of 1,25(OH)2D3 hold promise. As reviewed by Maestro and Seoane (2022) [201], 1778 ligands that specifically bind to the VDR receptor have been described so far, and the progress in understanding of their structure–function relationships has enabled modification of the structure of 1,25(OH)2D3 and also towards low-calcemic analogs. However, biological activities of these compounds have mainly been tested in the context of their anti-cancer or anti-psoriatic properties, and studying their potential neuroprotective effects has been neglected. Nevertheless, Grimm et al. (2017) [113] evaluated the effect of clinically used analogues of Vitamin D (maxacalcitol, calcipotriol, alfacalcidol, paricalcitol, doxercalciferol) on AD-relevant mechanisms and found that calcipotriol and maxacalcitol showed the strongest effect on Aβ-degradation in neuroblastoma cells or vitamin-D-deficient mouse brains. Additionally, calcipotriol was shown to suppress calcium-dependent α-synuclein aggregation by inducing calbindin-D28k expression, which may be relevant to the treatment of PD [202]. Overall, the VDR-mediated vitamin D neuroprotective, anti-inflammatory and neurovascular mechanisms may predominantly contribute to its beneficial effects in age-related ND treatment and prevention.

## Figures and Tables

**Figure 1 cells-12-00660-f001:**
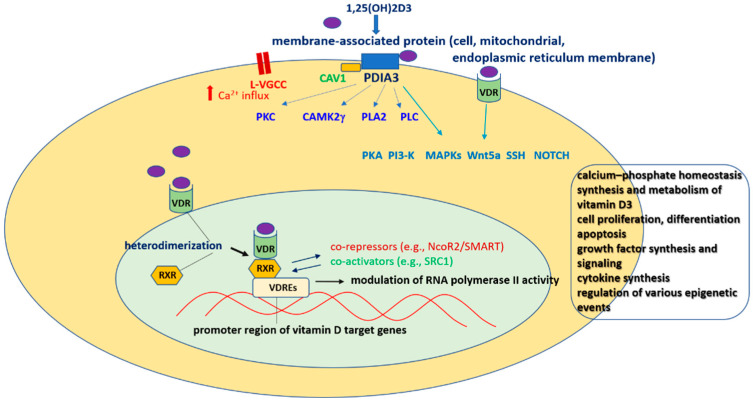
Vitamin D3 binding sites and signaling pathways. 1,25(OH)2D3—active form of vitamin D3; CAMK2g—g subunit of calcium/calmodulin-dependent protein kinase 2; CAV-1—caveolin-1; L-VGCC—L-type voltage-gated calcium channel; MAPK—Mitogen-Activated Protein Kinase; PDIA3—protein disulfide isomerase family A member 3; PI3-K—Phosphatidylinositol 3-Kinase; PKA—Protein Kinase A; PKC—Protein Kinase C; PLA2—Phospholipase A2; PLC—Phospholipase C; RXR—retinoid X receptors; SSH—sonic hedgehog; Wnt5a—ligand activating β-catenin-independent pathways; VDR—vitamin D receptor; VDRE—vitamin D response element.

**Figure 2 cells-12-00660-f002:**
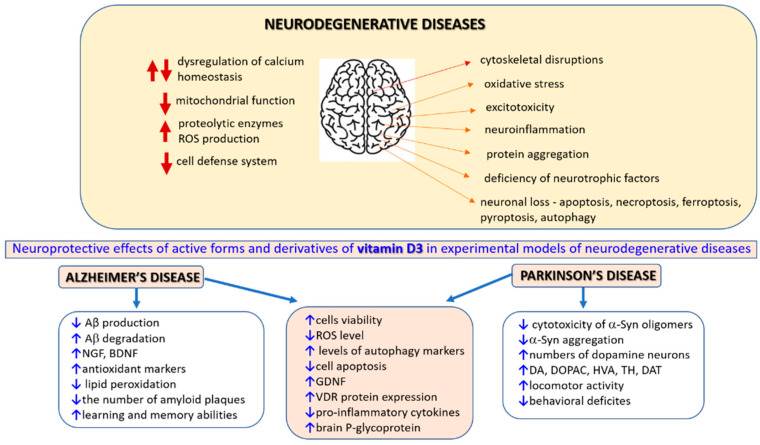
The pathogenic mechanisms of neurodegenerative diseases and neuroprotective effects of different forms of vitamin D3 in experimental models of Alzheimer’s and Parkinson’s diseases. Ab—amyloid b; BDNF—brain-derived neurotrophic factor; DA—dopamine; DAT—dopamine transporter; DOPAC—3,4-dihydroxyphenylacetic acid; GDNF—glial-derived neurotrophic factor; HVA—homovanillic acid; NGF—nerve growth factor; ROS—reactive oxygen species; Syn—synuclein; TH—tyrosine kinase; VDR— the vitamin D receptor.

**Table 3 cells-12-00660-t003:** Effects of vitamin D in experimental models of Parkinson’s disease—in vitro studies.

PD Models	Vitamin D Administration	Effects	References
Glutamate, 6-OHDA and MPP^+^-induced toxicity in rat mesencephalic culture	calcitriol (10–100 nM)	↑ cell viability	[162]
H_2_O_2_ and 6-OHDA-induced damage of primary cultures of rat ventral mesencephalon	calcitriol (0.1 n)/on 7DIV/before toxins	↑ cell viability	[163]
BSO and MPP^+^-induced neurotoxicity in rat cultured mesencephalic neurons	calcitriol (1–100 nM)	↑ neuron survival and neurite extension, ↓ ROS and gluthatione depletion	[164]
L-DOPA-induced neural stem cells (NSCs) injury	calcitriol (10–1000 nM)	↓ free radicals, ↑ cell viability and proliferation, ↑ prosurvival signaling, including activation of the PI3K pathway, and reducing oxidative stress	[165]
Rotenone-induced neurotoxicity in SH-SY5Y cells	calcitriol (2.5–10 μM)	↓ reactive oxygen species levels, ↑ levels of intracellular signaling proteins associated with cell survival; ↑ levels of autophagy markers (LC3, beclin-1, and AMPK)	[166]
Rat primary fetal ventral mesencephalic cultures of dopamine neurons	calcitriol (100 pM-100 nM)/7 days	↑ numbers of rat primary dopamine neurons ↑ GDNF expression ↓ dopamine neurons apoptosis	[167]
α-Syn-induced aggregation in SH-SY5Y cells	α-Syn oligomers +vitamin D (4 μM)/36 h	↑ cell viability ↓ cytotoxicity of α-syn oligomers ↓ α-Syn aggregation ↓ ROS	[168]

AMPK-5′AMP—activated protein kinase; BSO—buthionine sulfoximine; GDNF—glial cell-line-derived neurotrophic factor; LC3—microtubule-associated protein 1A/1B-light chain 3; MPP^+^-1—methyl-4-phenylpyridinium; 6-OHDA—6-hydroxydopamine; PD—Parkinson’s Disease; ROS—reactive oxygen species; α-syn—α-synuclein.

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
