# Peer review of "The Vitamin D Receptor as a Potential Target for the Treatment of Age-Related Neurodegenerative Diseases Such as Alzheimer’s and Parkinson’s Diseases: A Narrative Review"

_cells, 2023, doi:10.3390/cells12040660_

Round 1

Reviewer 1 Report

The manuscript summarises current studies on vitamin D receptor as a potential target for the treatment of age-related neurodegenerative diseases. The review is fairly comprehensive with some insightful comments. It has touched upon quite a few aspects on the subject, although some sections need major revisions. This reviewer has specifically the following comments: 

1. Ageing is a known risk factor for all neurodegenerative diseases.  The discussion of vitamin D from the perspective of "aging" is somewhat missing. Authors are advised to present a whole picture of the interconnected factors that refer to aging and neuronal damage in age-related neurodegeneration.

2. Several sections of the manuscript are lengthy and lack of readability. For example, the description of specificity of the commercially available antibodies in "Expression of the VDR in the brain" section is too much in detail. It is advisable to summarize but not to list literatures in tables.

3. Ref 109 in Table 2 seems to suggest that diet could lower the serum level of Vitamin D. This is not right. Vitamin D deficiency may not be the cause, but rather an early feature or an outcome of AD. I would like to see more discussion on how Vitamin D supplementation is associated with the risk of dementia.

4. Table 3 is too busy. The authors are encouraged to revise it by removing unnecessary details.

5. The argument that nVDR can be a potential target for neurodegenerative diseases is interesting although more evidence is needed. It would be appropriate to include reports on novel VDR ligands and differential involvement of VDRs in this regard.

Author Response

Ad. 1 Answer: Thank you for your insightful remark. We agree that the paper missed the important issue of vitamin D involvement in the basic mechanisms of aging processes. The revised paper has been supplemented with a new chapter “Brain aging” where some studies on brain aging and vitamin D were relocated from the chapter “Experimental studies in AD”.

Brain aging

Aging is a highly complex biological process, in which gradual impairment of antioxidant defense system, deterioration of immune system activity (immunosenescence), and decline in metabolic and regenerative processes take place. It has been postulated that the age-related increase in ROS generation, along with dysfunctional cellular repair or degradation mechanisms, can lead to elevated level of oxidized proteins and formation of amyloid (Squier, 2001). Moreover, it has been suggested that the pathogenesis of NDs, such as AD and PD is associated with the well-recognized hallmarks of aging: cellular senescence, loss of proteostasis, deregulated nutrient sensing, mitochondrial dysfunction, telomere attrition, genomic instability, epigenetic alterations, altered intercellular communication and stem cell exhaustion (Hou et al., 2019). Recently proposed new hallmarks of aging include autophagy, microbiome disturbance, altered mechanical properties, splicing dysregulation, and inflammation (Schmauck-Medina et al., 2022). An accumulating evidence indicates that vitamin D via both its genomic and non-genomic actions may influence aging processes. The VDR stimulation increases neurotrophin gene expression and promotes prosurvival and anti-inflammatory signaling pathways within the CNS. Vitamin D and its derivatives have been evidenced to protect neuronal cells against a variety of toxic agents at nanomolar concentrations and to prevent premature aging. In conjunction with the pivotal role of oxidative stress in the pathomechanism of brain aging and neuro-degeneration, it was reported that the combined treatment with calcitriol and the powerful antioxidant - lipoic acid protected primary mouse astrocytes against hydrogen peroxide-induced apoptosis and iron accumulation-induced damage (Molinari et al., 2019). A recent study demonstrated that the VDR is involved in attenuating ischemia-induced oxidative stress and brain injury via reciprocal activation SMAD family member 3 (SMAD3) and VDR transcription factors [77]. Berridge (2017) put forward a hypothesis that vitamin D due to its pleiotropic mechanisms including its effects on calcium homeostasis, controls the rate of aging and that its deficiency may increase the probability of age-related diseases, including PD and AD. Indeed, some preclinical data indicate that vitamin D3 could affect the biological hallmarks of aging. Thus, the administration of the food supplement containing Centella asiatica (L.) extract, vitamin C, zinc and cholecalciferol for 3 months restored telomerase reverse transcriptase expression and enhanced telomerase activity in the cerebellum and cortex of 18-month-old rats (Tsoukalas et al. 2021). Of note, calcitriol has been shown to stabilize 53BP1 and promote DNA double-strand break repair through the inhibition of CTSL, the up-regulation of which is a hallmark of premature aging (Gonzalez-Suarez et al., 2011). As reviewed by Ashapkin et al. ( 2019), there are some experimental evidences that vitamin D3 reduces progerin production and ameliorates progerin-related premature aging symptoms. In addition, an interaction of vitamin D with the ”aging suppressor” – protein Klotho has been considered. Klotho shows pleiotropic biological effects and its insufficiency appears to play a role in premature aging, whereas its overexpression promotes longevity. Specifically, deficiency in the vitamin D/Klotho/Nrf2 regulatory network may promote the age-related cognition deficits in rats, which can be attenuated by vitamin D supplementation (Berridge, 2016). Klotho is expressed in the brain and besides serving a plethora of biological functions it also regulates vitamin D metabolism (Dërmaku-Sopjani et al., 2021). Tuohimaa conducted studies on FGF23-/- and Klotho-/- mice and concluded that aging shows a U-shaped dependency on vitamin D levels, and that an optimal concentration of calcidiol to a higher extent than calcitriol is required to delay the aging process (Tuohimaa, 2009). It has been demonstrated that VDR KO mice showed premature aging features, such as alopecia, thickened skin and epidermal cysts, although no changes in the number of Purkinje cells in the cerebellum were found. Moreover, it has been shown that the expression of genes encoding NF-kB, Fgf-23, p53 and IGF1R was significantly lower in old VDR KO mice as compared to the controls. The authors pointed out that mice with D3 hypervitaminosis and aged VDR knockout mice are phenotypically similar and that genetic ablation of VDR promotes premature aging (Keisala et al., 2009). Messing et al. (2013) showed for the first time that exposure to vitamin D3 increased the lifespan of wild-type Caenorhabditis elegans, which expresses DAF-12 nuclear hormone receptor homologous to the vitamin D receptor in humans. This finding was confirmed by Mark et al. (2016) who also showed in a mechanistic study that cholecalciferol improved protein homeostasis and extended the lifespan of these worms via stress response pathway genes skn-1, ire-1, and xbp-1, and prevented protein aggregation and β-amyloid-induced toxicity. Vitamin D3 via immune regulation may also slow down the aging process. Thus, in aged mice vitamin D3 administration for 6 weeks reduced hallmarks of aging, such as retinal inflammation, retinal macrophage numbers, and amyloid beta accumulation (Lee et al., 2012). Other investigators suggested that a combination of calcitriol and mesozeaxathin could be a useful approach to contrast pathological features of Age-Related Macular Degeneration (AMD), such as retinal inflammation and oxidative stress (Lazzara et al., 2021). The anti-inflammatory effects of vitamin D3 administration in the retina may be relevant to therapies of neurodegenerative diseases because visual abnormalities due to AMD and glaucoma are often associated with AD and may occur before cognitive deficits (Ashok et al., 2020). Additional evidence has been provided that vitamin D3 can act as an anti-inflammatory agent by reducing the age-related microglial activation and increase in IL-1beta level in the rat hippocampus (Moore et al., 2005). Collectively, preclinical data suggest that vitamin D3 interferes with key regulators of aging processes and that a lack of VDR promotes progeria. However, the optimal vitamin D serum concentration for slowing down brain aging has not been established, yet. Peng et al. (2021)[100] reported the increase in Aβ accumulation in the mouse brain during aging which was associated with the n P-glycoprotein (P-gp) level. Calcitriol acting via VDR prevented age-related changes in P-gp level and partially reduced Aβ brain aggregates. Khairy and Attia (2021) investigated the possible protective effects of oral supplementation of cholecalciferol in young, middle-age and old rats and found that vitamin D significantly mitigated the aging-related reduction in brain BDNF (brain-derived neurotrophic factor) level and activities of AChE (acetylcholinesterase) and antioxidant enzymes, and reduced malondialdehyde (MDA) level and caspase-3 activity. Since they found a positive correlation between serum 25(OH)D level and brain BDNF or AChE activity and a negative correlation between serum vitamin D level and brain MDA levels or caspase-3 activity, they concluded that increasing brain BDNF could be a key mechanism of vitamin D anti-aging effect at least in the rat brain [Khairy and Attia, 2021]. The progression of the brain aging process which is often associated with dysfunctional hippocampal neurogenesis has been hypothesized to be accelerated by vitamin D deficiency [Gómez-Oliva et al., 2020]. In such scenario, vitamin D insufficiency in the senescent brain might cause changes in the proper function of Wnt signaling, which subsequently could lead to the failure in the control of the neurogenic homeostatic mechanisms involved in the protection of neural stem cells and in this way, could evoke cognitive impairments. A better understanding of the functional link between vitamin D, neurogenesis and cognitive performance during aging will create space for designing new therapies for age-related cognitive decline [Gómez-Oliva et al., 2020].

Ad. 2 Answer: 

We shortened some parts of the manuscript in order to make it more readable, especially the one discussing specificity of the commercially available antibodies. Now this fragment of the text reads:

“Wang et al. (2012) in their review indicated that early ELISA and immunohistochemical studies showed undetectable levels of VDR in the cerebrum and cerebellum in the rat and human, but also pointed out that the antibodies used for a direct measurement of VDR were not VDR-specific [71]. Recent imunohistochemical studies showed the presence of VDR in the cortex, caudate putamen, amygdala, reticular thalamus, and to a lesser extent in the hypothalamus, hippocampus, dorsal raphe nucleus, paraventricular thalamic nuclei, and bed nucleus of the stria terminalis in the brain of VDRCre mouse line [Liu et al., 2021].”

We decided to keep a list of papers dealing with experimental in vitro and in vivo data in Tables 1, 2, 4 and 5, since in these tables, the experimental details and investigated mechanisms are mentioned whereas in the text, they are described in a more general way. Moreover, we improved these tables by adding, whenever stated in original papers, which form of vitamin D was tested. Since in the chapters dedicated to clinical studies in AD and PD the studies were described in a considerable detail, in the revised manuscript we have decided to remove Table 3 and Table 6.  

Ad. 3 Answer: 

Thank you for your important comment. Indeed, a majority of studies showed that vitamin D status and vitamin D supplementation has a beneficial influence on brain health, and that in older patients, vitamin D supplementation either ameliorates or has no effect on declining cognitive function. We agree that the study by Lai et al. (2022) has serious limitations and may be misleading.

Therefore appropriate changes in the revised manuscript have been made:

Page 12. The fragment from the original version of the manuscript:  „Furthermore, supplementation with vitamin D increased Aβ deposition and exacerbated cognitive impairments. Thus, vitamin D supplementation in this animal model of AD did not reinstate the genomic VDR/RXR heterodimer but stimulated formation of the non-genomic VDR/p53 complex in brain tissue [109]. This intriguing hypothesis, which suggests the VDR/p53 signaling pathway as a possible target for neuroprotective drugs in AD, deserves further studies” has been replaced in the revised manuscript by the following one:

 „Furthermore, supplementation with vitamin D increased Aβ deposition and exacerbated cognitive impairments. They concluded that vitamin D supplementation in this animal model of AD did not reinstate the genomic VDR/RXR heterodimer but stimulated the formation of the non-genomic VDR/p53 complex in brain tissue [109]. However, a scientific rigor of this study has been questioned by Gombart et al., (2023) and Vieth (2022). In brief, the criticism of the Lai et al. (2022) report is focused on insufficient in vitro evidence, very high non-physiological concentrations of vitamin D, a limited value of the AD model in mice, and not adequate discussion of result of this study in the context of the prior literature.” Accordingly, a question mark was put into Table 2 ref. 109.

Page 17. The fragment from original version of the manuscript:  ”In contrast, a population-based longitudinal study by Lai et al. (2022) revealed that vitamin D3 supplementation increased the probability of developing dementia in older adults and increased the risk of mortality in patients with pre-existing dementia. This cohort study cautions against long-term use of vitamin D by AD patients [109].” has been replaced in the revised paper by the following one:

”In contrast, Lai et al. (2022) concluded based on a retrospective, population-based longitudinal study that vitamin D3 supplementation increased the probability of developing dementia in older adults and increased the risk of mortality in patients with pre-existing dementia [109]. However, an interpretation of these data appears to be quite inaccurate because the participants were treated with calcitriol, not as a dietary supplement, but as a strong medication prescribed to dementia-predisposed patients. It was strongly emphasized that the study by Lai et al. (2022) had serious methodological and interpretative limitations, confuses the drug commonly used for patients with kidney failure, with the nutritional use of vitamin D, and, therefore, does not justify the conclusions that “vitamin D supplementation worsens Alzheimer's progression” (Gombart et al.,(2023) and Vieth (2022).”

Ad. 4 Answer: Since in the chapters dedicated to clinical studies in AD and PD, the studies were described in considerable detail, in the revised manuscript we have decided to remove Table 3 and Table 6. 

Ad. 5 Answer: 

Indeed, there is still a great interest in designing novel VDR ligands. For example, novel des-D-ring interphenylene analogs of vitamin D3, have been recently designed and some of them showed high gene transcription promoting activities comparable to natural 1α,25(OH)2D3. Especially, the analogs having a meta-substituted side chain display high potency to form the complex with VDR and co-activator (Ibe et al., 2022). However, biological properties of these analogs have not been described, yet. Regarding therapeutic potential of these compounds in the treatment of neurodegenerative diseases, non-calcemic analogs of 1,25(OH)2D3 hold promise. As reviewed by Maestro and Seoane (Nutrients, 2022) 1778 ligands that specifically bind to the VDR receptor have been described so far, and the progress in understanding  of structure–function relationships enabled to modify the structure of 1,25(OH)2D3 also towards low-calcemic analogs. However, biological activities of these compounds are mainly tested in the context of their anti-cancer or anti-psoriatic properties and studying their potential neuroprotective effects is neglected. Nevertheless, Grimm et al. (2017)  evaluated the effect of clinically used analogues of Vitamin D (maxacalcitol, calcipotriol, alfacalcidol, paricalcitol, doxercalciferol) on AD-relevant mechanisms and found that calcipotriol and maxacalcitol, showed the strongest effect on Aβ-degradation in neuroblastoma cells or vitamin D deficient mouse brains. Also calcipotriol was shown to suppress calcium-dependent α-synuclein aggregation by inducing calbindin-D28k expression, which may be relevant in the treatment of PD (Rcom-H'cheo-Gauthier et al. 2017). These studies have been discussed in the manuscript.

Reviewer 2 Report

Dear Authors,

Your manuscript entitled "The nuclear vitamin D3 receptor (nVDR) as a potential target for the treatment of age-related neurodegenerative diseases such as Alzheimer's and Parkinson's diseases: A narrative review." is well structured and written. However, some minor issues should be considered before publication.

First, in order to be more specific, the authors should specify the differences between "vitamin D3" and "vitamin D", and should use them consequently. For e.g., in the Introduction, Line 97 or Line 222 "vitamin D3" refers to calcitriol, but in Table 1 "vitamin D3" and "1,25(OH)2D3" are used in parallel, without specifying whether they are synonyms or not.

In Lines 228-230, the authors used "vitamin D" without further details.  

On the other hand, at Lines 298-303 "vitamin D3" should mean cholecalciferol, not the active form, the 1,25(OH)2D3.

After all, the reviewer suggests that the most specific terminology should be used when referring to vitamin D3 metabolites (i.e., cholecalciferol, calcidiol, calcitriol, etc.). 

Minor suggestions:

Line 98-109. You should briefly introduce the nongenomic mechanisms and add appropiate references also (see 10.3390/nu13113672 and 10.3390/biomedicines10081824).

Figures should be considered to be added, in order to improve the readability of the manuscript.

Author Response

Thank you for your insightful remarks.

Answer for 1-4: The revised manuscript was improved in regard to used terminology for vitamin D compounds. Additionally in chapter 2 was added fragment containing basic information on vitamin D forms, synthesis and metabolism which familiarize the reader with used naming for vitamin D-originating compounds. In Tables were specified the used in particular studies vitamin D forms.

       We have added the following fragment on page 3:

Vitamin D is a fat-soluble substance belonging to the secosteroid family and its five forms (D1, D2, D3, D4 and D5) have been described to date. In humans, the main source (approx. 80%) of vitamin D derives from the skin where under sun exposure 7-dehydrocholesterol is transformed to unstable pre-vitamin D3 which further isomerizes to vitamin D3 (cholecalciferol). This form of vitamin D could also be consumed with animal-based food, whereas in plants, vitamin D2 (ergocalciferol) is present [55, Gáll and Székely, 2021]. The third source of vitamin D includes commercially available supplements in which cholecalciferol predominates as the compound of better bioavailability and biological activity when compared to ergocalciferol. To achieve their biological effects, these inactive forms of vitamin D in the first step need to be metabolized in the liver by CYP2R1 (25-hydroxylase) where 25-hydroxyvitamin D3 (25(OH)D3, calcifediol) or 25-hydroxyvitamin D2 (25(OH)D2) are produced. These circulating vitamin D metabolites are further hydroxylated by CYP27B1 (1 α-hydroxylase) present in target organs (e.g., the kidney) which produces 1α,25-dihydroxyvitamin D3 (1,25(OH)2D3, calcitriol) or 1α,25-dihydroxyvitamin D2 (1,25(OH)2D2), the active forms of vitamin D3 and D2, respectively. Since calcitriol maintains calcium and phosphate bone homeostasis, its production and metabolism in the kidney are precisely regulated. Only about 0.4% of active or circulating forms of vitamin D metabolites are available in plasma in free form whereas 58% are associated with vitamin D binding proteins (DBPs) which are highly polymorphic in humans causing interpersonal differences in vitamin D bioavailability. When in the body, the level of active and circulating (storage) forms of vitamin D is too high, especially after excessive supplementation, they are metabolized by CYP24A1 to inactive metabolites (1,24,25(OH)3D, calcitroic acid and 24,25(OH)2D).

Ad. 5 Answer: 

The non-genomic mechanism has been briefly described in Chapter 2 and the reference 10.3390/nu13113672 (review paper), pointed out by the reviewer, has been cited in this chapter. The second pointed Ref.  10.3390/biomedicines10081824 is an original paper and seems to be more relevant to the section “AD - experimental studies”, where it has been added and briefly described.

Added fragment on page 4:

“Rapid action of active vitamin D metabolites could be mediated by cell membrane-localized VDR which could modulate the activity of various signaling pathways (e.g., WNT, sonic hedgehog (Shh) and NOTCH signaling) or other membrane receptors (e.g., calcium channels or mitochondrial permeability transition pore). The enzyme PDIA3 (protein disulfide isomerase family A member 3, ERp57, 1,25D3-MARRS) is the best described target responsible for non-genomic action of active vitamin D metabolites [63]. By this mechanisms, they could regulate extracellular Ca2+ influx through L-VGCC, activated various protein kinases (e.g., CaMKII, PKC, PKA, PI3-K or MAPKs) and phospolipases (PLA2, PLC). Moreover, after binding with membrane receptors (VDR or PDIA3), active vitamin D metabolites could interact with some transcription factors (e.g. STAT3, NF-κB, Nrf2, RORα, RORγ, AhR) and in this way, they indirectly influence the expression of various genes. In contrast to genomic action of vitamin D via VDR, its non-genomic mechanisms are less recognized and mostly evidenced in in vitro settings [55,63].”

Ad. 6  Answer: 

In order to improve the readability of the manuscript we have added two figures. The Fig. 1 illustrates the VDR-mediated and non-genomic neuroprotective effects of Vitamin D, emphasizing the influence of Vitamin D on pro-survival biochemical pathways. The Fig. 2 shows neuroprotective effects of vitamin D in experimental models of neurodegenerative diseases.

Round 2

Reviewer 1 Report

The revised version has addressed all my concerns.

Author Response

Thank you for reading the revised version of our manuscript and acceptance of the changes introduced in the text.